# 🌉 Language-Pretraining-Induced Bias: A Strong Foundation for General Vision Tasks

**Yaxin Luo**                                                                  *Yaxin.Luo@mbzuai.ac.ae*
*Department of Machine Learning, MBZUAI*

**Zhiqiang Shen**                                                        *Zhiqiang.Shen@mbzuai.ac.ae*
*Department of Machine Learning, MBZUAI*

**Reviewed on OpenReview:** *https://openreview.net/forum?id=N7DSUbnzYo*

## Abstract

The ratio of "outlier" parameters in language pre-training models and vision pre-training models differs significantly, making **cross-modality** (language and vision) inherently more challenging than **cross-domain** adaptation[1]. As a result, many prior studies have focused on cross-domain transfer rather than attempting to bridge language and vision modalities, assuming that language pre-trained models are unsuitable for downstream visual tasks due to disparate parameter spaces. Contrary to this assumption, we show that adding a "bridge training 🌉" stage as a modality adaptation learner can effectively align Large Language Model (LLM) parameters with vision tasks. Specifically, we propose a simple yet powerful solution *random label bridge training* that requires no manual labeling and helps LLM parameters adapt to vision foundation tasks. Moreover, our findings reveal that partial bridge training is often advantageous, as certain layers in LLMs exhibit strong foundational properties that remain beneficial even without fine-tuning for visual tasks. This surprising discovery opens up new avenues for leveraging language pre-trained parameters directly within vision models and highlights the potential of partial bridge training as a practical pathway to cross-modality adaptation.

## 1 Introduction

Recent advancements in large language models [1, 49, 50, 56] have shown their remarkable capacity for capturing both syntactic and semantic information in an unsupervised manner. However, despite the rich internal representations these models develop, the utility of language-pretrained parameters in domains beyond text remains largely unexplored. Particularly in computer vision, while cross-domain adaptation (e.g., transforming a model trained on natural images to work on cartoon images) has received attention [53, 21, 55], true cross-modality transfer, i.e., from language to vision, presents a more substantial challenge. The fundamental discrepancy in input modalities (text tokens versus image pixels) suggests that LLM parameters might be inherently incompatible with the convolutional or transformer-based

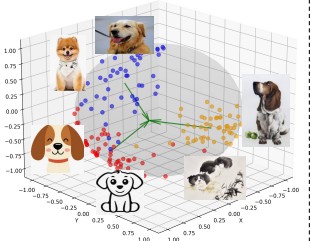 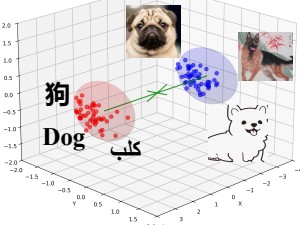

Cross Domain Adaptation          Cross Modality Adaptation

Figure 1: In cross-domain adaptation, the data type remains the same, though domains may vary in style or distribution. In contrast, cross-modality adaptation involves fundamentally different feature spaces, alongside variations in style or distribution.

---

[1]Cross-domain refers to different data distributions on the same modality, such as the adaptation from natural images to cartoon images for the same visual modality, rather than studying cross-modality (language modality and vision modality). Cross-modality is usually much more difficult than cross-domain as induced bias.

operations typical in visual tasks. As illustrated in Fig. 1, cross-domain adaptation modifies the distribution within the same modality, whereas cross-modality adaptation must reconfigure parameters trained on text tokens to handle image pixels.

One line of thought posits that the parameter spaces for language-based and vision-based models are simply too different for overlapping fine-tuning approaches. This has led many researchers to either ignore cross-modality transfer entirely or rely on cascading strategies where a vision model's output is fed into a language model like MLLMs scheme [2, 31, 9, 26] rather than overlapping fine-tuning them. While this division has simplified experimental design, it has also limited the exploration of potential synergies between language and vision. For instance, the ability of LLMs to capture high-level abstractions might help address challenges in dense vision tasks like object detection.

In many specialized domains such as industrial inspection, medical imaging and remote sensing, large-scale text corpora are readily available (manuals, logs, reports, protocols), while labeled images are scarce due to annotation cost and expertise requirements. In this regime, it is valuable to reuse language-pretrained parameters as a prior and adapt them to visual inputs with minimal annotation: we can first perform an annotation-free bridge stage on unlabeled images, and then fine-tune with the small labeled image set for the downstream task.

While direct cross-modality parameter transfer is often regarded as nontrivial in prior transfer or adaptation settings [5, 13, 45, 17, 39, 61], our findings show that a properly designed "bridge training" stage can effectively adapt language-pretrained parameters for visual tasks. This is also compatible with recent evidence that representation alignment can emerge across diverse models and modalities [20]. Specifically, we propose a *random label bridge training* protocol: we assign random labels to visual data and optimize the model using a standard supervised objective, requiring no human annotation. Although the labels carry no semantic information, this stage encourages the network to learn a broad mapping from image structure to internal representations, thereby reshaping the parameter statistics and reducing the modality gap between text and images. Importantly, random label training is flexible and label-free: it does not require a curated labeled dataset, enabling rapid cross-modality adaptation without the constraints of supervised annotation.

A further important and surprising discovery in our work is that *partial bridge training* can sometimes yield even better results than full-layer adaptation. Certain layers in large language models appear to contain fundamental representational capacities that remain beneficial for vision tasks. By freezing these layers when training, we preserve their internal structures and exploit the fact that some of the linguistic abstractions can also be advantageous for visual processing. Through extensive empirical analysis, we find that retaining these *frozen* layers not only reduces the number of trainable parameters but also preserves beneficial biases that have been distilled through language pre-training.

Our results demonstrate consistent gains over various baselines that do not employ language-pretrained parameters. Specifically, random label bridge training achieves +11.5% and +14.1% accuracy improvements on CIFAR-10/100 under linear probing, leveraging language bias further boosts these gains to +19.6% and +21.3%. Moreover, empirical studies show that random label training leads to more stable parameter alignment than naive direct fine-tuning, highlighting the non-triviality of cross-modality adaptation.

Our contributions are threefold: (1) We challenge that, while architectures are shared across modalities and cross-modal alignment has been observed, it remains nontrivial and not guaranteed that language-pretrained weights can be directly repurposed for vision under weak supervision [28, 33, 35], we investigate this empirically and theoretically; (2) We propose and validate a parameter alignment approach *random label bridge training* that is entirely annotation-free; and (3) We demonstrate that partial bridge training can be sufficient to preserve the valuable representations learned by large language models, thus laying the groundwork for broader research into cross-modality transfer.

## 2 Related Work

**Cross-Domain and Cross-Modality Adaptation.** Adapting models to new domains or modalities is a fundamental challenge in machine learning. *Cross-domain adaptation* transfers models within same modality and addresses distributional shifts using techniques like domain adversarial training [10, 38], domain

Figure 2: **Outlier parameters and weight distributions in models trained on different modalities.** Language-pretrained GPT-2 shows a markedly heavier-tailed distribution with numerous large-magnitude "Outlier" weights, whereas vision-pretrained ViT and GPT-2 structure model trained on images exhibit fewer outliers and narrower spreads.

alignment [40, 60], test-time training [48, 51], and self-supervised learning [18, 15, 8]. While effective for tasks such as natural-to-synthetic image transfer, these methods are less suited for bridging different modalities. *Cross-modality adaptation*, such as between language and vision, introduces challenges due to structural and distributional differences. Vision-language models (VLMs) like CLIP [44] and Flamingo [3], and Multimodal LLMs [9, 29], leverage paired datasets to learn joint representations but rely heavily on aligned supervision and task-specific designs. In contrast, our work explores inherent cross-modality capabilities of LLMs, leveraging pretraining-induced biases for annotation-free, flexible adaptation without architectural changes.

**Modality Difference in Pretraining.** Pretraining on language and image data poses challenges due to differences in input structures and parameter distributions.

Existing multimodal approaches address this differences via learning rate modulation [41, 27, 23], representative embeddings [12, 57], or alternating unimodal training [58, 62, 54], but primarily focus on joint training setups. Our work leverages pretraining-induced biases in LLMs for vision tasks through random label bridge training, aligning parameter distributions while retaining beneficial linguistic abstractions. This approach offers a practical solution to modality difference without requiring joint training.

**Connection to representation alignment.** Recent work suggests that representation spaces across architectures, objectives, and even modalities may become increasingly aligned as models scale, motivating the view that cross-modal representational similarity can emerge as a general phenomenon (e.g., The Platonic Representation Hypothesis [20]). Building on this perspective, our focus is not to assert that language and vision are *incompatible* but to empirically characterize when and how cross-modality transfer can be induced by a simple training procedure. In particular, we analyze the dynamics and outcomes of a random-label bridge stage (which removes semantic label supervision) and layer-wise/partial updating (which isolates where adaptation is most effective), providing both theoretical insight and controlled experiments on cross-modality adaptation.

## 3 *"Outlier"* Parameters in Foundation Models

**"Outlier" parameters in language pre-training models and vision pre-training models.** A key observation in our study is that large language models exhibit a higher proportion of "outlier" parameters compared to models pre-trained on visual data. Intuitively, this stems from the inherent difference between discrete textual tokens and continuous image signals. In language tasks, each token typically maps to a sparse, high-dimensional embedding space, which can lead to large fluctuations or "spikes" in certain parameter values during training. As a result, the parameter distribution in LLMs tends to have heavier tails, reflecting a higher likelihood of extreme values. Conversely, in visual models, continuous signals such as pixel intensities provide smoother gradients and more uniformly distributed training examples, thereby reducing the tendency for large, isolated parameter magnitudes.

Our statistical analysis in Fig. 2 supports these theoretical considerations. By comparing magnitude distributions of weights in multiple layers, we find that LLMs contain a significantly greater percentage of high-magnitude parameters and exhibit a heavier-tailed distribution overall. Visual models, on the other hand, show relatively flatter distributions with fewer large outliers. These empirical findings validate the notion that

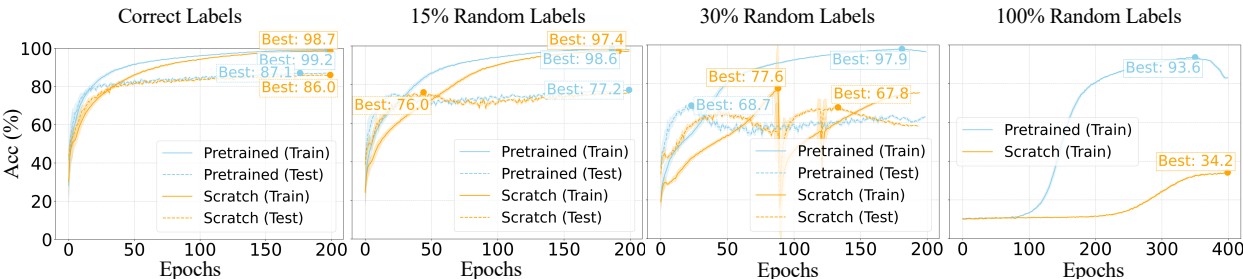

Figure 3: Train and test acc. curves for pretrained vs. scratch GPT-2 on CIFAR-10 using varying random label proportions (0%, 15%, 30%, 100%). Pretrained models consistently show higher accuracy and faster convergence, indicating enhanced robustness to label noise.

discrete tokens induce more extreme parameter values and show practical implications for model initialization, optimization stability, and potential transfer learning strategies when bridging language and vision tasks. In the following, we provide a formal definition of cross-modality adaptation and outline our problem setup, to explain the approach we use to mitigate the mismatch between these fundamentally different parameter spaces.

**Definition 3.1** (Cross-Modality Adaptation). *Consider a source data distribution $\mathcal{S}$ (source modality) and a target data distribution $\mathcal{T}$ (target modality), where $\mathcal{S}$ is defined over $\mathcal{X}_S \times \mathcal{Y}_S$ and $\mathcal{T}$ is defined over $\mathcal{X}_T \times \mathcal{Y}_T$. Let $\mathbf{X}_\mathcal{S} \times Y_\mathcal{S}$ be the input and output spaces associated with $\mathcal{S}$, and $\mathbf{X}_\mathcal{T} \times Y_\mathcal{T}$ be the input and output spaces associated with $\mathcal{T}$. We define a learning task $t_S$ as the expected risk minimization problem induced by $(S, \ell_S)$: learn a predictor $f_S : \mathcal{X}_S \to \mathcal{Y}_S$ by minimizing $\mathcal{R}_S(f) = \mathbb{E}_{(x,y) \sim S}\big[\ell_S(f(x), y)\big]$, and similarly define $t_T$ (and $f_T$) induced by $(T, \ell_T)$. Here, "derived from $\mathcal{Y}_S$" means that the prediction target and loss are defined on the output space $\mathcal{Y}_S$ (together with the conditional $\mathcal{S}(Y \mid X)$). The goal of cross-modality adaptation is to enhance the target predictive function $f_\mathcal{T}$ for the target task $t_\mathcal{T}$ by leveraging knowledge obtained from the source $\mathcal{S}$ and $t_\mathcal{S}$, provided that $\mathcal{S} \neq \mathcal{T}$. Here, $\mathcal{S} \neq \mathcal{T}$ implies both a difference in their marginal input distributions $\mathcal{S}_\mathbf{X} \neq \mathcal{T}_\mathbf{X}$ (i.e., $\mathbf{X}_\mathcal{S} \neq \mathbf{X}_\mathcal{T}$ and $\mathcal{S}_\mathbf{X}(\mathbf{X}) \neq \mathcal{T}_\mathbf{X}(\mathbf{X})$ ) or a difference in the tasks themselves $t_S \neq t_T$ (i.e., $Y_S \neq Y_\mathcal{T}$ or $\mathcal{S}(Y \mid \mathbf{X}) \neq \mathcal{T}(Y \mid \mathbf{X})$ ).*

## 4 Empirical Discoveries on Language Induced Bias for General Vision Tasks

It may appear counterintuitive [6] to leverage weights from one modality (e.g., language) to benefit another (e.g., vision). However, in this section we provide both theoretical and empirical evidence that language-pretrained parameters can indeed help visual tasks. Specifically, Section 4.1 demonstrates how a pretrained language bias accelerates convergence and improves performance on image classification. Section 4.2 then introduces random label training may potentially be a good robustness trainer and proves (see Theorem 4.1) how training on unlabeled visual data aligns these language-driven parameters to the image distribution. Section 4.3 analyzes loss landscapes, showing how pretraining smooths optimization and improves feature discriminativeness. Finally, Section 4.4 demonstrates that language-induced bias enables models to learn separable visual features even with random labels, reinforcing its role as a strong unsupervised regularizer.

### 4.1 Pretrained Language Accelerates Convergence & Improve Performance for Vision Tasks

To empirically investigate the optimization effects and benefits of language-pretrained LLM weights for vision tasks, we compare the performance of pretrained GPT-2 against models trained from scratch on image classification tasks. Specifically, we evaluate both approaches on CIFAR-10, CIFAR-100 and ImageNet-1k datasets under identical training conditions. In our image adaptation, the first layer is a newly introduced patch embedding/input projection (randomly initialized); pretrained GPT-2 weights are used only in subsequent transformer blocks. As shown in Table 1 and Fig. 3 (leftmost plot), the pretrained models converge more quickly and reach higher accuracies than their scratch-trained counterparts. Specifically, on CIFAR-10, the Pretrained-GPT-2 model gains +0.5% Top-1 accuracy in training, while on CIFAR-100 it improves test accuracy by +2.4%. Similar patterns hold for ImageNet-1K, where pretrained models consistently obtain better test performance. These findings demonstrate that language-induced bias facilitates optimization on

Table 1: Comparison of language-pretrained GPT-2 weights and scratch-initialized GPT-2 models on CIFAR-10, CIFAR-100 and ImageNet-1K for image classification. (**Note**: we only train GPT-2 on ImageNet-1k for 150 Epochs and LLAMA3.2-1B for 50 Epochs because of resource limitations.)

| Model | CIFAR-10 | | CIFAR-100 | | ImageNet-1K | |
|---|---|---|---|---|---|---|
| | Train-acc1 | Test-acc1 | Train-acc1 | Test-acc1 | Train-acc1 | Test-acc1 |
| Scratch-GPT-2 | 98.7 | 86.0 | 98.8 | 58.0 | 56.4 | 66.5 |
| Pretrained-GPT-2 | $99.2_{+0.5}$ | $87.1_{+1.1}$ | $99.1_{+0.3}$ | $60.4_{+2.4}$ | $59.2_{+2.8}$ | $68.8_{+2.3}$ |
| Scratch-LLAMA3.2-1B | 99.3 | 79.7 | 99.7 | 49.3 | 66.3 | 64.0 |
| Pretrained-LLAMA3.2-1B | $99.6_{+0.3}$ | $89.4_{+9.7}$ | $99.8_{+0.1}$ | $64.7_{+15.4}$ | $73.8_{+7.5}$ | $67.7_{+3.7}$ |

Table 2: Performance comparison of language-pretrained and scratch-initialized GPT-2 models under varying ratios of random labels on CIFAR-10 and CIFAR-100. Darker " ■ " and " ■ " indicates higher Train-acc1 and Test-acc1, respectively.

| Model | Random Ratio | CIFAR-10 | | CIFAR-100 | |
|---|---|---|---|---|---|
| | | Train-acc1 | Test-acc1 | Train-acc1 | Test-acc1 |
| Scratch-GPT-2 | 15% | 97.4 | 76.0 | 98.5 | 45.2 |
| | 30% | 75.4 | 67.8 | 98.2 | 32.5 |
| | 100% | 34.2 | 2.5 | 65.9 | 0.9 |
| Pretrained-GPT-2 | 15% | $98.6_{+1.2}$ | $77.2_{+1.2}$ | $99.0_{+0.5}$ | $48.1_{+2.9}$ |
| | 30% | $97.9_{+22.5}$ | $68.7_{+0.9}$ | $98.9_{+0.7}$ | $33.7_{+1.2}$ |
| | 100% | $93.9_{+59.7}$ | $6.3_{+3.8}$ | $98.7_{+32.8}$ | $0.8_{-0.1}$ |

vision modality by guiding the model toward better solutions. This not only accelerates convergence but also results in improved performance on vision tasks.

**Better Fitting Ability.** Fig. 3 further compares accuracy across three different fractions of the random labels: 15%, 30%, and the entire random (100%). In each scenario, the model initialized with GPT pre-trained weights achieves notably higher training accuracy than a baseline trained from scratch, despite having the same architecture. What makes this observation particularly intriguing is that the labels are completely random, yet the GPT-pretrained model significantly exhibits a stronger ability to fit the data but scratch-trained model is close to collapse. This phenomenon reflects that language pre-training furnishes an effective inductive bias, one that remains useful even when the task's labels have no inherent structure.

### 4.2 Is Random Label Tuning A Good Robustness Trainer and Adapter to Target Modality?

As analyzed in [34], it has been shown analytically for convolutional and fully connected networks that an alignment occurs between the principal components of the network parameters and the data. However, our focus is on Transformer-based LLM architectures like GPT [43]. While there are shared theoretical insights between our work and prior studies regarding covariance matrix alignment between network parameters and data, our approach diverges significantly. Specifically, we present that principle of random label training can be formalized as:

**Theorem 4.1** (Random Label Training Induces Covariance Alignment). *Suppose $x$ are drawn i.i.d. from a distribution with covariance $\Sigma_x$, and the initial weights $w$ in the first layer are drawn from an isotropic (e.g., Gaussian) distribution. Then when we train a neural network on these inputs $x$ using random labels (under typical conditions such as SGD training), the learned covariance $\Sigma_w := \mathbb{E}[ww^T]$ of the first-layer weights aligns with the data covariance $\Sigma_x$. Concretely, $\Sigma_w$ shares eigenvectors with $\Sigma_x$, and the map from each eigenvalue $\sigma_i^2$ of $\Sigma_x$ to the corresponding eigenvalue $\tau_i^2$ of $\Sigma_w$ follows a well-defined (experimentally smooth) transfer function $f(\cdot)$.*

**Intuition and key ideas.** Theorem 4.1 is aligned with prior analyses of random-label training; our novelty is the covariance-alignment formulation, spectral transfer function perspective and direct connection to our bridge-stage method. (i) *Isotropy and symmetry*: Initially drawing $w$ from an isotropic distribution means there is no preferred direction in parameter space. During random-label training, every label is equally "wrong", so the only structure that can be exploited or "learned" is in the input data $x$. (ii) *Eigenvector*

*alignment*: By gradient descent dynamics (or SGD), directions in $w$-space that correspond to larger variance directions in $x$-space receive larger updates on average. Over the course of training, despite the labels being random, the weight vectors align with the most significant eigenspaces of $\Sigma_x$. As a result, the principal components of $\Sigma_w$ converge to those of $\Sigma_x$. (iii) *Eigenvalue mapping*: Each eigenvalue $\sigma_i^2$ of $\Sigma_x$ gets mapped to a corresponding $\tau_i^2$ for $\Sigma_{w.}$. Empirically, one finds a smooth function $f(\sigma)$ such that $\tau_i \approx f(\sigma_i)$. Typically, larger input variances $\sigma_i^2$ lead to higher effective learning rates, increasing the corresponding $\tau_i^2$ before backpropagation balances out other directions in the network. (iv) *Implication for "real" labels*: Surprisingly, even when labels are random, the network still *adapts* its first-layer weights to the input structure. Once this alignment occurs, subsequent fine-tuning or re-training with real labels can proceed more efficiently, as the first-layer filter directions already match the major modes of variation in the data. Importantly, we emphasize that unlike prior method [34], random label training is uniquely feasible within our approach. Furthermore, self-supervised or TTT methods, such as entropy minimization [52], could also be incorporated into our framework.

We evaluated the robustness of language-pretrained LLM weights under noisy conditions using varying random label ratios (15%, 30%, and 100%) on CIFAR-10 and CIFAR-100. As shown in Table 2 and Fig. 3, pretrained models consistently outperformed scratch-initialized ones across all conditions. For instance, with 15% random labels, pretrained models achieved 77.2% and 48.1% test accuracies on CIFAR-10 and CIFAR-100, respectively, compared to 76.0% and 45.2% for scratch models. Even with 100% random labels, pretrained models demonstrated significantly better structured learning, achieving 93.9% training accuracy on CIFAR-10 versus 34.2% for scratch models. These results underscore the robustness of language-pretraining-induced biases as a powerful regularizer for extracting meaningful patterns without semantic labels.

### 4.3 Language Bias Induces Better Optimized Loss Landscapes

Although Theorem 4.1 demonstrates how random-label training can induce first-layer weight alignment under simplified conditions (e.g., single-layer settings and controlled data distributions), real-world large models and complex natural-image data introduce additional challenges. To bridge this gap, we investigate a deeper architecture (GPT-2) on actual image data, again leveraging random labels. Fig. 4 reveals that the scratch model has a rugged landscape in both cases, indicating many local minima that can hinder optimization. In contrast, the pretrained GPT-2 starts from a very different point: although the landscape is not necessarily smoother overall, quantitative analysis in Table 15 shows it begins near a high-energy saddle point, which is easier to escape during training. Specifically, we observe a sharper eigenvalue decay rate (–11.6 vs. –3.7), a higher Hessian trace (305k vs. 21k), and a much larger spectral gap (207k vs. 226), all suggesting stronger curvature in select directions [14]. Yet, most directions remain flat, as indicated by a lower participation ratio (0.34 vs. 0.78), and the model shows lower noise sensitivity and gradient predictiveness [36]. The pretrained model also has a higher ratio of negative eigenvalues (0.65 vs. 0.35) and greater sensitivity to certain parameters. Together, these findings imply that language pretraining positions the model in a landscape that facilitates optimization by guiding it out of poor local minima and toward more stable, generalizable solutions.

### 4.4 Language-Pretraining-Induced Bias Helps Models Learn Separable Features without Annotations

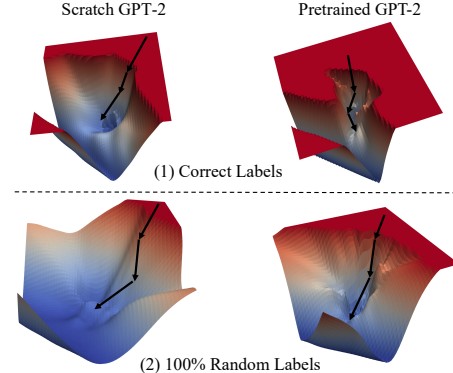

Figure 4: **Loss landscape on CIFAR-10.** We visualize a 2D cross-section of the high-dimensional loss surface by plotting $L(\theta_0 + \alpha d_1 + \beta d_2)$, where $d_1 = \theta_T - \theta_0$ is the training direction and $d_2$ is a random direction orthogonal to $d_1$ (both per-layer normalized). The **X-axis** and **Y-axis** correspond to the coefficients $\alpha$ and $\beta$, respectively (height/color indicates loss). *Top*: Correct Labels Training. *Bottom*: 100% Random Labels Training. *Left*: Scratch GPT-2. *Right*: Pretrained GPT-2.

Beyond the robustness observations in Section 4.2, we further investigate the representational differences between pretrained and scratch-initialized GPT-2 models when trained with 100% random labels on CIFAR-10.

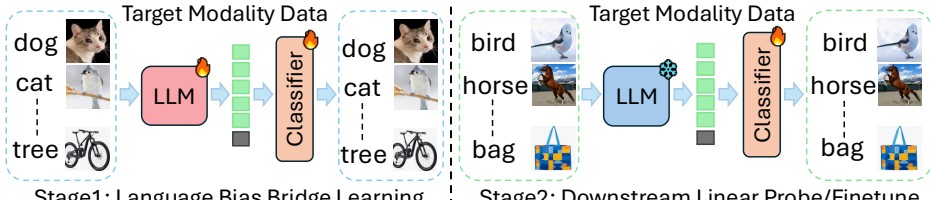

Figure 6: **Overview of our two-stage Language Bias Bridge Learning framework.** In Stage 1, the pretrained LLM is adapted to the target modality under random labels, In Stage 2, a lightweight classifier refines these representations on real labels.

Concretely, we extract the final-layer hidden states $\mathbf{h}_i \in \mathbb{R}^d$ for each sample $i$, where $d$ denotes the hidden dimension. We then apply a t-SNE mapping $f_{\text{t-SNE}} : \mathbb{R}^d \to \mathbb{R}^2$ and perform KMeans clustering on the reduced features. Formally, each sample $\mathbf{h}_i$ is assigned to a cluster $\mathbf{c}_i$ via:

$$\mathbf{c}_i = \arg\min_k \big\| f_{\text{t-SNE}}(\mathbf{h}_i) - \mu_k \big\|^2, \quad \forall i \in \{\text{samples}\}, \tag{1}$$

where $\mu_k$ is the centroid of cluster $k$.

As illustrated in Fig. 5, pretrained GPT-2 model's embeddings become more separable (top right), forming tight intra-class clusters even under completely random labels, while scratch model exhibits less structure. This indicates a powerful cross-modal transfer effect: random-label procedure still leverages the language-induced parameter bias, enabling the model to uncover latent image structures. In contrast, scratch model lacks such an inductive bias, yielding minimal improvement. These results corroborate our earlier findings that language pretraining serves as a strong regularizer under noisy conditions, leading to faster convergence and better unsupervised discriminative capabilities in visual domain.

## 5 Language Bias Bridge Training as A Modality Adaptation Learner

### 5.1 Overall Proposed Formulation

Figure 5: t-SNE embeddings for pretrained (top) and scratch-initialized (bottom) GPT-2 models on CIFAR-10 under 100% random labels. *Left*: Before training, both appear partly disordered. *Right*: After training, the pretrained model achieves far tighter, more coherent clusters, indicating superior feature separability compared to the scratch model.

Building on the insights from Section 3, we propose a **Language Bias Bridge Training (LBBT, 🌉)** paradigm, designed to leverage language-pretraining-induced bias for Cross-Modality adaptation. In particular, we exploit a 100% random-label setting on the target modality to train a large language model (LLM) for vision tasks, followed by a lightweight adaptation stage for downstream domains. Fig. 6 illustrates the proposed two-stage framework.

Let $\mathcal{S} = (\mathbf{X}_\mathcal{S}, Y_\mathcal{S})$ be the *source modality*, where $\mathbf{X}_\mathcal{S} = \{x_i\}_{i=1}^N$ and $Y_\mathcal{S} = \{\tilde{y}_i\}_{i=1}^N$ denote inputs and random labels, respectively. Let $\mathcal{T} = (\mathbf{X}_\mathcal{T}, Y_\mathcal{T})$ be the *target modality* with inputs $\mathbf{X}_\mathcal{T} = \{x_j\}_{j=1}^M$ and true labels $Y_\mathcal{T} = \{y_j\}_{j=1}^M$. We assume $\mathcal{S}_\mathbf{X} \neq \mathcal{T}_\mathbf{X}$ and $t_\mathcal{S} \neq t_\mathcal{T}$, reflecting discrepancies in either the input distributions or the tasks themselves. Our goal is to learn a representation function $\text{LLM}(\cdot; \theta)$ that transfers knowledge from $\mathcal{S}$ to $\mathcal{T}$, improving the predictive function $f_\mathcal{T}$ for the target task. We also train a lightweight adapter $A(\cdot; \phi)$ to refine the LLM representations for downstream use on $\mathcal{T}$. Formally, we minimize:

$$\min_{\theta,\phi} \Big[ \underbrace{\mathcal{L}_{\text{bridge}}\big(\mathcal{S}_\mathbf{X}, \mathcal{S}_Y; \text{LLM}(\cdot; \theta)\big)}_{\text{Stage 1: 🌉}} + \underbrace{\mathcal{L}_{\text{adapt}}\big(\mathcal{T}_\mathbf{X}, \mathcal{T}_Y; \text{LLM}(\cdot; \theta), A(\cdot; \phi)\big)}_{\text{Stage 2: Downstream Adaptation}} \Big], \tag{2}$$

where $\mathcal{L}_{\mathrm{bridge}}$ is a supervised loss on random-labeled data (Stage 1), and $\mathcal{L}_{\mathrm{adapt}}$ is a supervised or semi-supervised loss on real-labeled data (Stage 2). The bridge stage aligns the pretrained LLM with the target modality without requiring semantic labels, and the downstream stage then fine-tunes these representations with actual task supervision.

## 5.2 Random Label Bridge Training

The foundation of our approach lies in the inherent biases induced by language pretraining, which, as demonstrated in Section 4.2, serve as a powerful regularizer under noisy conditions. To fully leverage these biases for cross-modality adaptation, we introduce LBBT that aligns the feature representations of LLMs with vision tasks, even though the labels we use are random and offer no semantic guidance. Training a model on 100% random labels typically seems infeasible because there is no meaningful supervisory signal. However, we hypothesize and empirically confirm that language-pretrained weights encode robust feature hierarchies that can still be refined through a supervised objective, even when the labels themselves are random. Rather than relying on semantically correct annotations, the model learns to harness the structural properties of the input data, guided by its strong, language-induced initialization. Formally, let $S = (X_S, Y_S)$ denote our source modality with inputs $X_S$ and random labels $Y_S$. Let $\mathrm{LLM}(x; \theta)$ represent the feature embedding of input $x$ under parameters $\theta$. Our bridge learning objective is then given by:

$$\mathcal{L}_{\mathrm{bridge}}(S; \theta) \; = \; \frac{1}{|X_S|} \sum_{x \in X_S} \ell_{\mathrm{RSL}}\big(\mathrm{LLM}(x; \theta), Y_S\big), \tag{3}$$

where $\ell_{\mathrm{RSL}}$ is a supervised loss function (e.g., cross-entropy) computed against the random labels. Despite offering no true semantic structure, these labels force the model to refine its features based on the patterns present in $X_S$, effectively bridging the domain gap from language to vision.

As shown in Section 4.2, pretrained LLM weights exhibit notable robustness even under extreme noise such as 100% random labels due to the biases acquired during language pretraining. These biases naturally organize the feature space, promoting better separability and coherence. We preserve and further shape these beneficial biases by treating bridge training as a supervised task with random labels. During LBBT, the LLM parameters $\theta$ adapt to the statistical properties of visual data, thereby transferring language-induced structures into effective visual representations even in the absence of meaningful labels.

**Target Modality Downstream Adaptation** Following the initial alignment achieved through LBBT, the second stage refines representations for specific target tasks. This involves training a lightweight downstream adapter $A(\cdot; \phi)$ with task-specific labeled data while optionally fine-tuning pretrained LLM parameters $\theta$. The supervised loss function for this stage is:

$$\mathcal{L}_{\mathrm{adapt}}(T; \theta, \phi) = \frac{1}{|X_T|} \sum_{(x,y) \in T} \ell_{\mathrm{CE}}\left(A(\mathrm{LLM}(x; \theta); \phi), y\right), \tag{4}$$

where $\ell_{\mathrm{CE}}$ is cross-entropy loss. This stage leverages language-pretraining-induced biases to accelerate convergence and improve generalization, completing an two-step framework for cross-modality transfer.

**Extend to partial bridge training**. Prior work shows that neural networks memorize random labels largely through their later layers [34], supported by evidence from classification accuracy on layer activations and intrinsic dimensionality estimates [4]. Inspired by these findings, we hypothesize selectively freezing certain layers during LBBT preserves inductive biases formed in earlier layers of pretrained LLMs. Early layers shaped by language pretraining capture broad structural features are readily transferable across modalities. Updating only layers most prone to task-specific memorization can improve cross-modality adaptation.

Our experiments confirm this hypothesis (Fig. 8). Surprisingly, training only the first five layers of the LLM during Language Bias Bridge Training yields faster convergence and better downstream performance compared to updating all layers. This underscores the critical role of early layers in maintaining robust, transferable representations. Moreover, results suggest tuning every layer can weaken beneficial semantic structures acquired from language pretraining, ultimately hindering generalization. In summary, partial bridge training offers an effective and computationally efficient strategy by focusing on the layers most

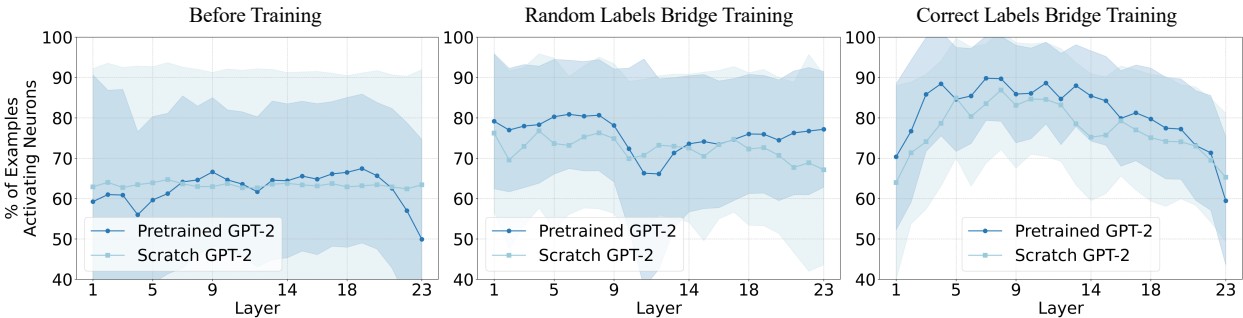

Figure 7: **Layer-wise neuron activation ratios for image samples in GPT-2 models.** For each block $\ell$, we record the MLP post-GELU activations $\mathbf{A}^{(\ell)}(x) \in \mathbb{R}^{T \times d_{\mathrm{mlp}}}$ on real images, compute token-averaged neuron responses, and count neuron $k$ as activated by image $x$ if $\bar{a}_k^{(\ell)}(x) > 0$. *Left*: before training. *Middle*: random-label bridge training. *Right*: correct-label bridge training. Training increases $r_\ell$, indicating greater utilization of MLP capacity for visual inputs.

pertinent to modality-specific adjustments. This approach reduces training overhead while leveraging the inherent strengths of pretrained LLMs, making them more adaptable to diverse downstream tasks.

## 6 Final Empirical Analysis

**Dataset and Implementation Details**. Experiments are conducted on CIFAR-10, CIFAR-100 [24], TinyImagenet-200 [25], and ImageNet-1K [11] using GPT-2-medium [42] with 355M parameters. For Bridge Training, the model is trained for 400 epochs on CIFAR-10 (batch size 64, single L20 GPU) and 100 epochs on ImageNet-1K (batch size 32, 6×4090 GPUs). Linear probing experiments use a batch size of 128 for 100 epochs on an L20 GPU. And all the sacle up experiments for Qwen3-8B, Qwen3-14B and LLaVA series are all conducted on 8×A100 GPUs. All experiments use a consistent training setup with 224×224 input resolution, a learning rate of 1e-3 (Adam optimizer, cosine annealing), weight decay of 0.05, drop path regularization [19] of 0.1, and gradient clipping with max norm 1.0.

**Hyperparameter selection and robustness.** We clarify how hyperparameters are chosen and evaluate sensitivity to optimization choices. We acknowledge that learning rate and weight decay can affect absolute performance; thus, we perform a small learning-rate × weight-decay sweep. To avoid unintentionally favoring LBBT, we select a single hyperparameter setting using a method-agnostic criterion on the baseline w/o LBBT (e.g., held-out validation or best-performing baseline setting), and then *freeze* this choice and apply the same hyperparameters to LBBT and all other comparisons. We summarize the sensitivity results in Table 3. While absolute accuracy varies across lr/wd, LBBT consistently improves over w/o LBBT across all evaluated settings. For example, under lr = 1e−3 and wd = 0.05, LBBT improves the CIFAR-100 linear-probe test accuracy from 17.4% (w/o LBBT) to 35.0% (w/ LBBT), i.e., a +17.6% absolute gain.

Table 3: **Hyperparameter sensitivity study.** We sweep learning rate (lr) and weight decay (wd) under the same training protocol and report linear-probe test acc1 on CIFAR-100 (%). Each entry is reported as *w/o LBBT / LBBT*; we additionally report the absolute gain as subscript.

|            | wd = 0.05        | wd = 0.1          | wd = 0.2          |
|------------|------------------|-------------------|-------------------|
| lr = 5e−4  | 15.1 / 31.0$_{+15.9}$ | 10.5 / 30.3$_{+19.8}$ | 10.3 / 23.5$_{+13.2}$ |
| lr = 3e−4  | 17.2 / 32.4$_{+15.2}$ | 13.9 / 31.1$_{+17.2}$ | 12.6 / 24.8$_{+12.2}$ |
| lr = 1e−3  | 17.4 / 35.0$_{+17.6}$ | 16.6 / 33.3$_{+16.7}$ | 13.7 / 26.8$_{+13.1}$ |

**Neurons Activating Ratio by Images.** To quantify how much of the MLP capacity is utilized by visual inputs, we measure a layer-wise *neuron activation ratio* using real image samples. For each Transformer block $\ell$, we record the MLP hidden activations *after* the nonlinearity (GELU), denoted by $\mathbf{A}^{(\ell)}(x) \in \mathbb{R}^{T \times d_{\mathrm{mlp}}}$ for an image $x$, where $T$ is the number of visual tokens and $d_{\mathrm{mlp}}$ is the MLP hidden width. We aggregate per

neuron by token-averaging:

$$\bar{a}_k^{(\ell)}(x) = \frac{1}{T} \sum_{t=1}^{T} \mathbf{A}_{t,k}^{(\ell)}(x), \tag{5}$$

and count neuron $k$ as *activated* by image $x$ if $\bar{a}_k^{(\ell)}(x) > 0$. The layer-wise activation ratio is then

$$r_\ell = \frac{1}{N\, d_{\text{ff}}} \sum_{i=1}^{N} \sum_{k=1}^{d_{\text{ff}}} \mathbb{1}\left[\bar{a}_k^{(\ell)}(x_i) > 0\right], \tag{6}$$

i.e., the fraction of (image, neuron) pairs that are activated at layer $\ell$.

Fig. 7 compares the percentage of image samples that trigger neuron activations across GPT-2 layers in three stages: initialization, random-label bridge training, and correct-label bridge training. We observe a clear increase in activation ratios under both training methods, indicating the model devotes more neurons to processing visual inputs. Notably, the random-label bridge training yields higher activation rates in some layers and is more consistent with the correct-label counterpart. This suggests that random labels not only align the model's parameters for cross-modality adaptation but also make the model more sensitive to images, potentially even more so than correct-label supervision.

**Language Bias Bridge Training.** From Table 4, we compare Bridge Training (models weights are random initialized and trained from scratch) and Language Bias Bridge Training (load pretrained model weights with language data). It is evident that leveraging language-pretraining-induced bias significantly improves performance across diverse datasets. On CIFAR-10, Language Bias Bridge Training achieves a train accuracy of 93.9%, surpassing standard Bridge Training by +59.7%. Similar trends are observed on CIFAR-100 (+32.8%) and ImageNet-1K (+2.8%), highlighting the consistent advantage of incorporating language-induced biases. These results demonstrate the effectiveness of our approach in aligning LLM parameters with visual tasks, even under challenging scenarios with random labels. In addition, in Table 5, we extend the scale to 8B and 14B LLMs to show that the observed benifits can still hold in large-scale models.

Table 4: **Random-label bridge training across datasets of varying scale.** We report training top-1 accuracy (`Train-acc1`) on the *random* bridge labels. **Bridge Training** denotes training the same GPT-2 architecture on the visual dataset with 100% random labels, starting from *randomly initialized* weights. **Language Bias Bridge Training (LBBT)** uses the *identical* random-label bridge objective, data, and hyperparameters, but initializes from *language-pretrained* GPT-2 weights.

| Paradigm | CIFAR-10 | CIFAR-100 | TinyImageNet-200 | ImageNet-1K |
| | Train-acc1 | Train-acc1 | Train-acc1 | Train-acc1 |
| --- | --- | --- | --- | --- |
| Bridge Training | 34.2 | 65.9 | 94.2 | 53.7 |
| Language Bias Bridge Training | 93.9$_{+59.7}$ | 98.7$_{+32.8}$ | 95.3$_{+1.1}$ | 56.4$_{+2.8}$ |

Table 5: **LBBT's Benefits also hold when scale up to larger LLMs.**

| Paradigm | LLM Backbone | CIFAR-10 | CIFAR-100 | TinyImageNet-200 |
| | | Train-acc1 | Train-acc1 | Train-acc1 |
| --- | --- | --- | --- | --- |
| Bridge Training | Qwen3-8B | 88.2 | 84.2 | 90.0 |
| Language Bias Bridge Training | Qwen3-8B | 95.9$_{+7.7}$ | 93.3$_{+9.1}$ | 92.4$_{+2.4}$ |
| Bridge Training | Qwen3-14B | 91.6 | 88.3 | 91.7 |
| Language Bias Bridge Training | Qwen3-14B | 97.9$_{+6.3}$ | 97.7$_{+9.4}$ | 94.3$_{+3.4}$ |

**Downstream Linear Probe.** To assess the effectiveness of our bridge training paradigm in a cross-modality setting, we employ a linear probe to evaluate the quality of the learned representations. As shown in Table 14, a model trained without bridge training achieves 62.7% on CIFAR-10 and 35.0% on CIFAR-100. Incorporating the bridge training step boosts performance to +11.5% and +14.1%, demonstrating that aligning representations under random labels yields more discriminative features. Finally, adopting language bias bridge training further raises accuracies by +19.6% and +21.3%, indicating that leveraging the inherent bias in language-pretrained parameters provides an additional strong advantage for cross-modality adaptation.

Table 6: Object Detection Results on COCO 2017 Dataset.

| Model | AP | AP50 | AP75 | APs | APm | APL |
|-------|-----|------|------|-----|-----|-----|
| DETR | 36.2 | 58.0 | 38.4 | 15.0 | 39.1 | 53.8 |
| Language Bias DETR | $38.3_{+2.1}$ | $59.2_{+1.2}$ | $39.6_{+1.2}$ | $16.6_{+1.6}$ | $41.7_{+2.6}$ | $54.2_{+0.4}$ |

**Compare with other Self-Supervised Methods.** To address the concern that a scratch GPT-2 baseline may be non-standard for vision, we additionally compare LBBT with representative SSL objectives under a controlled protocol (ImageNet-100 pretraining with the same backbone and training budget, followed by identical linear probing). As shown in Table 7, LBBT achieves the strongest linear-probe test accuracy among the compared SSL objectives in this matched-budget setting. We also report DINO with a ViT-L/16 backbone as a strong vision-native SSL reference (not directly comparable due to different backbone/recipe), to contextualize the absolute performance gap between general-purpose language initialization and specialized vision SSL pretraining.

Table 7: Linear-probe image-classification accuracy on CIFAR-100. (all pretrained on imagenet100 first.)

| Method | Train-acc1 | Test-acc1 |
|--------|-----------|-----------|
| **Unstructured SSL Methods** | | |
| Language Bias Bridge Training | 21.8 | 33.4 |
| Rotation Image | 25.1 | 32.2 |
| **Structured SSL Methods** | | |
| MAE | 24.5 | 30.7 |
| SimCLR | 16.2 | 26.2 |
| **Standard vision SSL reference (different backbone/recipe)** | | |
| DINO (ViT-L/16) | – | $62.2^{\dagger}$ |

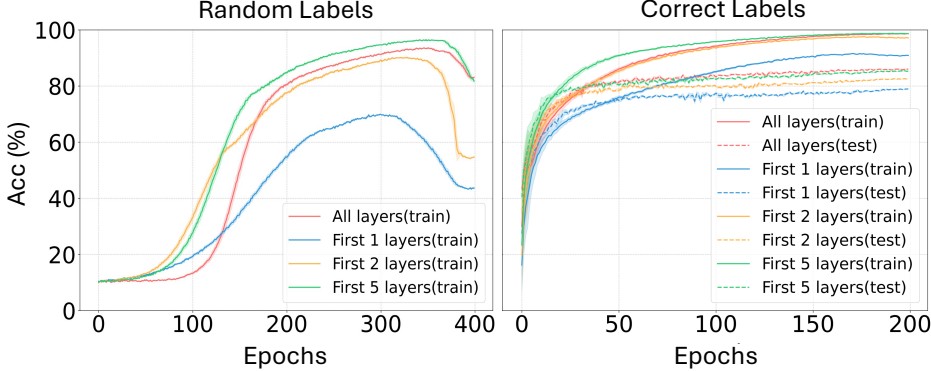

Figure 8: **Partial Bridge Training results under both random-label and correct-label settings.** Updating only the first 2 layers already matches the performance of training all layers, and training the first 5 layers surpasses it.

**Partial Bridge Training.** Fig. 8 compares training all layers versus only the early layers under both correct-label and random-label regimes. Surprisingly, restricting updates to just the first few layers can match or even outperform full-layer training. Specifically, Table 8 and Table 12 show that on CIFAR-10 with correct labels, training only the first 5 layers achieves 85.5%, close to 87.1% from training all layers. Under random labels, the gap is even more striking: updating only the first 5 layers yields +2.6% training accuracy than training all layers. A similar trend holds for CIFAR-100, confirming that early layers shaped by language pretraining are key to robust, transferable representations. By contrast, fully updating deeper layers can weaken these beneficial linguistic biases. Hence, partial bridge training reduces computational costs and preserves valuable language-induced structures, enabling strong cross-modality performance.

Table 8: **Partial Bridge Training.** Darker " ▇ " and " ▇ " indicates higher Train-acc1 and Test-acc1, respectively. Note that under random-label training, the test accuracy is naturally low because the labels have no true class semantics; hence, the training accuracy is the primary indicator of model's fitting ability.

| Layers | CIFAR-10 | | CIFAR-100 | |
|---|---|---|---|---|
| | Train-acc1 | Test-acc1 | Train-acc1 | Test-acc1 |
| *Correct Labels :* | | | | |
| All Layers | 99.2 | 87.1 | 99.1 | 60.4 |
| First 1 Layer | 90.2 | 73.9 | 88.5 | 49.4 |
| First 2 Layers | 97.5 | 82.6 | 97.9 | 54.7 |
| First 5 Layers | 98.9 | 85.5 | 99.3 | 59.7 |
| *Random Labels :* | | | | |
| All Layers | 93.9 | 6.3 | 98.7 | 0.8 |
| First 1 Layer | 69.9 | 4.3 | 77.7 | 1.3 |
| First 2 Layers | 90.1 | 7.2 | 95.7 | 1.2 |
| First 5 Layers | 96.5 | 6.7 | 99.2 | 0.9 |

**Extend to Vision Dense Prediction Tasks.** For dense prediction tasks, we *do not* directly use GPT-2 as a visual backbone. Instead, we leverage the architectural compatibility of Transformer blocks and *initialize* the Transformer modules in dense prediction models using pretrained GPT-2 weights. **DETR.** We follow the standard DETR pipeline with a convolutional backbone (e.g., ResNet) that produces a spatial feature map, which is flattened into a token sequence and processed by a Transformer encoder–decoder. For compatibility with GPT-2, we instantiate the DETR Transformer with the same hidden size / head count / FFN width and depth as the chosen GPT-2 checkpoint (e.g., $d$=768, 12 heads, FFN 3072, 12 encoder layers and 12 decoder layers). We then copy GPT-2 block parameters into each DETR Transformer layer: the self-attention projection weights, FFN weights, and LayerNorm parameters. Since GPT-2 does not contain a cross-attention module, we initialize the decoder cross-attention using GPT-2 attention weights as an initialization. All other DETR components (CNN backbone, input projections, object queries, and prediction heads) follow the original DETR design and are trained normally. The *Scratch* baseline uses the same DETR architecture but with random initialization. **Segmenter.** We follow Segmenter with a ViT encoder and a mask-transformer decoder. We keep the ViT encoder unchanged, and initialize the mask-transformer decoder blocks by copying pretrained GPT-2 block weights (attention/MLP/LayerNorm) for as many layers as available. Other decoder-specific parameters (e.g., class embeddings and projection layers) are initialized as in original Segmenter implementation. The *Scratch* baseline uses same Segmenter architecture without GPT-2 initialization.

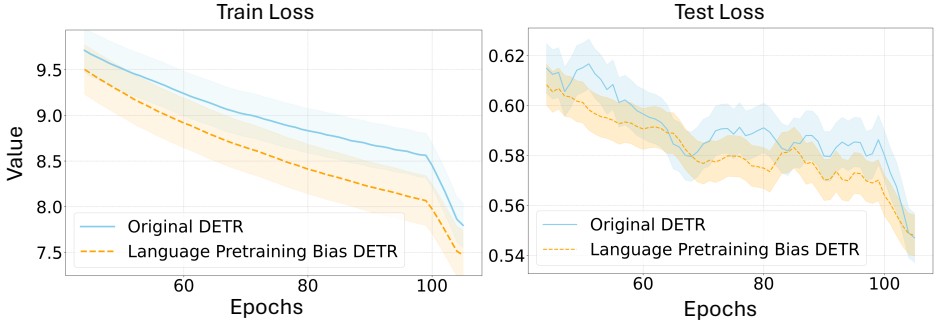

Figure 9: **Train and test loss curves for DETR.** Language-pretrained initialization accelerates convergence and lowers final loss, highlighting its benefit for dense vision tasks.

To further validate the effectiveness of language pretraining bias beyond classification, we evaluate its impact on dense prediction tasks, specifically object detection using DETR [7] on COCO2017 [30], semantic segmentation on AED20K [59], Pascal Context [37] with Segmenter [47] (Image Retrieval in Appendix Table 11). Table 6, Table 9 and Fig. 9 demonstrate the models initialized with language-pretrained weights exhibit better performance than training the model from scratch. This suggests that language pretraining-

Table 9: Semantic Segmentation Results on AED20K and Pascal Context.

| Model | AED20K | | Pascal | |
|---|---|---|---|---|
| | miou(ss) | miou(ms) | miou(ss) | miou(ms) |
| Segmenter-T-Mask/16 | 37.8 | 39.1 | 44.9 | 45.5 |
| Language Bias Segmenter-T-Mask/16 | $38.5_{+0.7}$ | $39.5_{+0.4}$ | $46.7_{+1.8}$ | $47.4_{+1.9}$ |

induced biases not only enhance representation learning for classification but also benefit structured vision tasks requiring fine-grained spatial reasoning.

**Apply Language Bias to Vision Encoder of Multimodal Large Language Model.** We further explore our hypothesis that initializing vision models with language-biased weights enhances cross-modal capabilities. To substantiate this claim, we conduct experiments under LLaVA-style training pipelines across different model scales (7B and 13B) and different LLM backbones (Vicuna and Mistral). Due to the complexity of directly comparing language-initialized models against randomly initialized ones in a fully end-to-end multimodal pretraining setup, we adopt an alternative experimental design. Specifically, we replace the default CLIP [44] ViT-based vision encoder in LLaVA [32] with a GPT-2 Transformer-based encoder. In our **Language Bias** setting, we initialize the GPT-2 vision encoder with weights pretrained on language tasks. Conversely, in the **Scratch** setting, the vision encoder is initialized randomly. Both configurations follow the identical training regimen and data used in the corresponding LLaVA recipes. Results in Table 10 show that language-pretrained initialization consistently improves performance over scratch initialization on multiple VQA and visual reasoning benchmarks, and the gains persist across model scales and LLM backbones.

Table 10: Comparison on three Multimodal VQA and one Reasoning Benchmarks across different model scales and LLM backbones. (*Pretrained rows are highlighted; subscript denotes absolute gain over the corresponding scratch setting.*)

| Model | LLM Backbone | SQA | TextVQA | GQA | MM-Vet |
|---|---|---|---|---|---|
| Scratch-gpt2-encoder-LLaVA1.5-7B | Vicuna-7B | 65.5 | 42.7 | 38.6 | 11.1 |
| Pretrained-gpt2-encoder-LLaVA1.5-7B | Vicuna-7B | $66.7_{+1.2}$ | $44.2_{+1.5}$ | $41.2_{+2.6}$ | $18.5_{+7.4}$ |
| Scratch-gpt2-encoder-LLaVA1.6-7B | Mistral-7B | 17.7 | 68.5 | 47.8 | 40.9 |
| Pretrained-gpt2-encoder-LLaVA1.6-7B | Mistral-7B | $21.0_{+3.3}$ | $69.1_{+0.6}$ | $49.3_{+1.5}$ | $43.2_{+2.3}$ |
| Scratch-gpt2-encoder-LLaVA1.6-13B | Vicuna-13B | 22.5 | 70.3 | 51.9 | 44.3 |
| Pretrained-gpt2-encoder-LLaVA1.6-13B | Vicuna-13B | $24.6_{+2.1}$ | $71.8_{+1.5}$ | $56.1_{+4.2}$ | $48.7_{+4.4}$ |

## 7 Conclusion

This work showed that language-pretrained parameters can serve as an effective prior for vision adaptation, improving both optimization dynamics and learned representations. Across multiple settings, a simple random-label bridge stage induces substantial alignment to visual inputs and improves downstream performance without semantic supervision. A learning-rate and weight-decay sensitivity study further shows that while absolute performance varies across hyperparameters, LBBT consistently improves over the corresponding w/o LBBT baseline under all evaluated settings. Partial training, updating only a small subset of early layers, retains useful pretrained structure, reduces adaptation cost, and can match full-layer updating in several regimes. We further validate transfer beyond image classification. Initializing Transformer components in dense prediction models with language-pretrained weights yields gains on object detection and semantic segmentation, suggesting the benefit extends to spatially structured tasks. In multimodal settings, initializing a GPT-2-based vision encoder in LLaVA-style pipelines improves performance across VQA and reasoning benchmarks, model scales, and LLM backbones. Overall, these results suggest cross-modality transfer from language to vision can emerge under weak supervision, motivating Language Bias Bridge Training (LBBT) as a practical, label-free, and compute-efficient framework for cross-modality adaptation. Future work includes scaling to larger language backbones, extending to other modalities, and combining random-label bridging with weak/self-supervised signals to further improve semantic alignment while keeping label cost low.

## Acknowledgements

This work is supported by the MBZUAI-WIS Joint Program for Artificial Intelligence Research.

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

## Appendix

In this appendix, we provide additional material that complements the main paper:

- **Section A: Detailed Proofs.** Formal derivations supporting Theorem 4.1, showing how random-label training induces first-layer alignment (see §4.1 of the main paper).

- **Section B: More Visualizations.** Additional qualitative figures that expose weight-distribution shifts and outlier emergence (Fig. 10, Fig. 11); complements the discussion in §3.

- **Section C: Results on Partial Training.** Layer-freeze ablations and learning-curve analysis under both correct and random labels (Tab. 8, Fig. 13); extends §6.

- **Section D: Fine-grained Layer-wise Outlier Analysis.** Heat-maps of neuron-level outliers across all 24 GPT-2 layers (Fig. 14); augments the main outlier study in §3.

- **Section E: Comparison with Time-Series Pretraining Bias.** Side-by-side evaluation of language- vs. time-series-pretrained weights on vision datasets (Tab. 13).

- **Section F: More Linear Probing Results on w/o Bridge Training and With Bridge Training Comparison.** (Tab. 14).

- **Section G: Quantitative Loss-Landscape Analysis.** Hessian statistics and stability metrics (Tab. 15); supports the landscape visualizations in §4.3.

## Limitation Statement

Our study focuses on standard image-classification (CIFAR-10/100, Tiny-ImageNet-200, ImageNet-1K) and dense-prediction tasks (COCO2017, ADE20K, Pascal Context), so although results suggest broad promise, confirming generality on additional modalities such as audio, video, and long-range vision–language reasoning remains future work; similarly, while we demonstrate the method with GPT-2-medium (355M parameters) and give preliminary evidence that larger backbones behave consistently (LLaMA3.2-1B), a systematic sweep across model scales and transformer variants is deferred to follow-up studies. Finally, because the annotation-free bridge stage relies on 100% random labels, tasks demanding fine-grained semantic alignment e.g, captioning could benefit from hybrid schemes that mix random and weak supervision, which we leave for future investigation.

## A   Detailed Proof

Below is the self-contained, more formal proof of our stated Theorem 4.1, drawing on the standard group-invariance argument to show why training with random labels can align first-layer parameters to the data covariance. Throughout, we assume the following:

**1. Input Assumptions.** Each input $x \in \mathbb{R}^d$ is drawn i.i.d. from a distribution with zero mean and covariance $\Sigma_x \in \mathbb{R}^{d \times d}$. For concreteness and simplicity, $x$ can be taken to be Gaussian (i.e., $x \sim \mathcal{N}(0, \Sigma_x)$, or more generally, drawn from a distribution $\mathcal{D}$ whose symmetry group is large enough to include all orthogonal transformations that preserve $\Sigma_x$.

**2. Network Assumptions.** The first layer of the neural network is either fully connected embedding or (a patch-based) convolution for image input. In either case, each first-layer weight vector $w \in \mathbb{R}^d$ interacts with $x$ primarily through inner products $\langle w, x \rangle$. The initial weights $w \in \mathbb{R}^d$ in the first layer are drawn i.i.d. from an isotropic distribution with mean zero and covariance $\sigma^2 I$. (For concreteness, $w \sim \mathcal{N}(0, \sigma^2 I)$.)

**3. Label Assumptions (Random Labels).** Each training instance $(x, y)$ has label $y$ drawn *independently and uniformly* from a finite label set $\{1, 2, \ldots, c\}$, *regardless* of $x$. This implies there is no genuine correlation between input $x$ and label $y$.

**4. Training Assumption.** We train the first-layer parameters (and possibly deeper layers) by stochastic gradient descent (SGD) for $T$ steps. The crucial point being random, the *only* structure the model sees is in $x$.

## Proof.

We adopt an invariance argument using the orthogonal group [22, 16, 34]:

$$\mathcal{G} \;=\; \big\{\, G \in O(d) \,\big|\, G^T \Sigma_x G \;=\; \Sigma_x \big\}, \tag{7}$$

where the set of all orthogonal matrices that leave $\Sigma_x$ invariant. The proof proceeds in two main steps:

1. Step 1 (Invariance in Distribution). We show that, at each iteration $t$, the distribution of $w$ remains invariant under the action of $\mathcal{G}$. That is, if $w \sim \mu_t$ is the distribution of weights at iteration $t$, then $G_w \sim \mu_t$ for all $G \in \mathcal{G}$.

2. Step 2 (Alignment of Covariances). Because the data distribution and the random labels provide no directional bias other than $\Sigma_x$ itself, the limiting covariance $\Sigma_w$ of $w$ must share the same eigenspaces as $\Sigma_x$. Concretely, any distribution $\mu$ that is invariant under all $G \in \mathcal{G}$ must be aligned with $\Sigma_x$. One then shows that each eigenvalue $\sigma_i^2$ of $\Sigma_x$ maps to a corresponding eigenvalue $\tau_i^2$ of $\Sigma_w$ via some function $f$.

We now detail these two steps:

**Step 1: Invariance in Distribution Under Orthogonal Transformations**

1. Definition of Group Action on $w$ and $x$.

Let $w \in \mathbb{R}^d$ and $x \in \mathbb{R}^d$. For each $G \in \mathcal{G} \subseteq O(d)$, define

$$(G \cdot w, G \cdot x) = (G_w, G_x), \tag{8}$$

Since $G \in O(d)$ preserves inner products, $\langle G_w, G_x \rangle = \langle w, x \rangle$.

2. Invariance of Data Distribution.

Because $G \in \mathcal{G}$ satisfies $G^T \Sigma_x G = \Sigma_x$, it leaves the distribution of $x$ invariant. Concretely, $x \sim \mathcal{N}(0, \Sigma_x)$ implies $Gx \sim \mathcal{N}(0, \Sigma_x)$ as well. Thus sampling $x$ and then applying $G$ yields a sample from the *same* distribution.

3. Initial Weights Are Isotropic.

By assumption, the initial weight distribution $w_0$ is isotropic: $\mathbb{E}\big[w_0 w_0^T\big] = \sigma^2 I$. Hence $G_{w_0} \sim w_0$. This implies that at $t = 0$, the distribution $\mu_0$ of $w_0$ is invariant under $\mathcal{G}$.

4. Loss Function Under Random Labels.

The training loss at iteration $t$ is

$$\mathcal{L}(w_t; x, y) \;=\; \ell\Big(\langle w_t, x \rangle,\, y\Big). \tag{9}$$

Since $y$ is random and uncorrelated with $x$, each gradient update

$$w_{t+1} = w_t - \eta \nabla w L(w_t; x, y)$$

depends only on the scalar product $\langle w_t, x \rangle$. Crucially, if $w_t$ and $x$ follow distributions that are invariant under group action by $\mathcal{G}$, the next update remains invariant as well.

5. SGD Update Commutes with Group Action.

Formally, one must show that for each $G \in \mathcal{G}$,

$$G_{w_{t+1}} \;=\; G\big(w_t - \eta\,\nabla w L(w_t;\, x,\, y)\big) \;=\; (G_{w_t}) - \eta\,\nabla G_{w_t} L(G_{w_t};\, G_x,\, y), \tag{10}$$

where the last equality holds because $\langle G_{w_t}, G_x \rangle = \langle w_t, x \rangle$ and the label $y$ is unchanged by $G$. By induction, this invariance holds at every iteration $t$. Thus the distribution $\mu_t$ of $w_t$ satisfies $G_{w_t} \sim w_t$ for all $G \in \mathcal{G}$.

**Step 2: Covariance Alignment Follows from Distribution Invariance**

We now argue that a distribution $\mu$ on $w \in \mathbb{R}^d$ invariant under $\mathcal{G}$ implies that its covariance $\Sigma_w = \mathbb{E}\left[ww^T\right]$ aligns with $\Sigma_x$. More precisely: 1. Spectral Decomposition of $\Sigma_x$.

Since $\Sigma_x \in \mathbb{R}^{d \times d}$ is symmetric positive semi-definite, we can decompose $\Sigma_x = K_1 \oplus K_2 \oplus \cdots \oplus K_r$ where each $K_i \subseteq \mathbb{R}^d$ is an eigenspace of $\Sigma_x$ corresponding to eigenvalue $\sigma_i^2$. By definition of $\mathcal{G}$, each $K_i$ is invariant under $\mathcal{G}$.

2. $\mathcal{G}$-Invariance Implies Each $K_i$ Is an Invariant Subspace of $\Sigma_w$.

Because $w \sim \mu$ is invariant under $\mathcal{G}$, for any $G \in \mathcal{G}$, we also have $w' \coloneqq G_w \sim \mu$. By taking expectations, one shows that if a vector subspace $K_i$ is $\mathcal{G}$-invariant for $\Sigma_x$, it must also be $\mathcal{G}$-invariant for $\Sigma_w$. Concretely, if $u \in K_i$, consider $G_u$ for all $G \in \mathcal{G}$. This forces $\Sigma_w$ to preserve $K_i$, otherwise $\Sigma_w$ would not be consistent with the distribution-level invariance.

3. Hence $\Sigma_w$ Shares Eigenspaces with $\Sigma_x$.

Since each $V_i$ is forced to be an invariant subspace of $\Sigma_w$, we conclude that $\Sigma_w$ and $\Sigma_x$ must diagonalize in the same basis. In other words, there exist scalars $\tau_i^2$ such that

$$\Sigma_w(v) = \tau_i^2 v, \quad \forall v \in V_i$$

This shows that $\Sigma_w$ and $\Sigma_z$ *share* eigenvectors but differ in their eigenvalues $\tau_i^2$ vs. $\sigma_i^2$.

4. Eigenvalue Mapping $\sigma_i^2 \mapsto \tau_i^2$.

Empirically, one observes a smooth function $f(\cdot)$ such that $\tau_i \approx f(\sigma_i)$. Intuitively, directions in $x$-space with larger variance $\sigma_i^2$ yield (via random-label SGD) more significant updates to $w$, driving the corresponding $\tau_i^2$ higher until other competing directions partially balance out. This effect is captured by a stable "transfer function" $f$, which can be measured experimentally by plotting $\tau_i$ vs. $\sigma_i$.

## B    More Visualizations

**Weight Distributions Expose Language Pretraining's Advantage for Visual Tasks.** In addition to the weight parameter distribution visualization (shown on the right) in Fig. 2, we also provide further comparisons between pretrained and from-scratch GPT-2 models trained on correct labels, as well as an analysis of these models' distributions under random-label bridge training. Fig. 10 compares the learned weight distributions of GPT-2 models trained from scratch on images (blue) versus pretrained on language data (orange), under both correct labels (left plot) and 100% random labels (right plot). When trained with correct labels, the scratch model's distribution remains narrowly peaked near zero, suggesting many weights do not significantly deviate from their initialization. In contrast, the pretrained model shows a broader spread of values, reflecting a more active reconfiguration of parameters that appears guided by its preexisting linguistic priors.

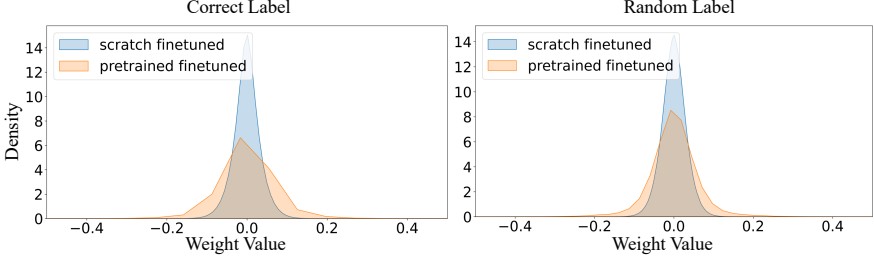

Figure 10: **Weight Parameter Distribution.** We compare weight distributions of GPT-2 models trained from scratch (*blue*) and with language-pretrained weights (*orange*) under correct labels (left) and 100% random labels (right). The pretrained model displays a smoother, more heavily tailed distribution, highlighting its ability to adapt to visual data—even with noisy, non-semantic supervision, thanks to latent linguistic priors.

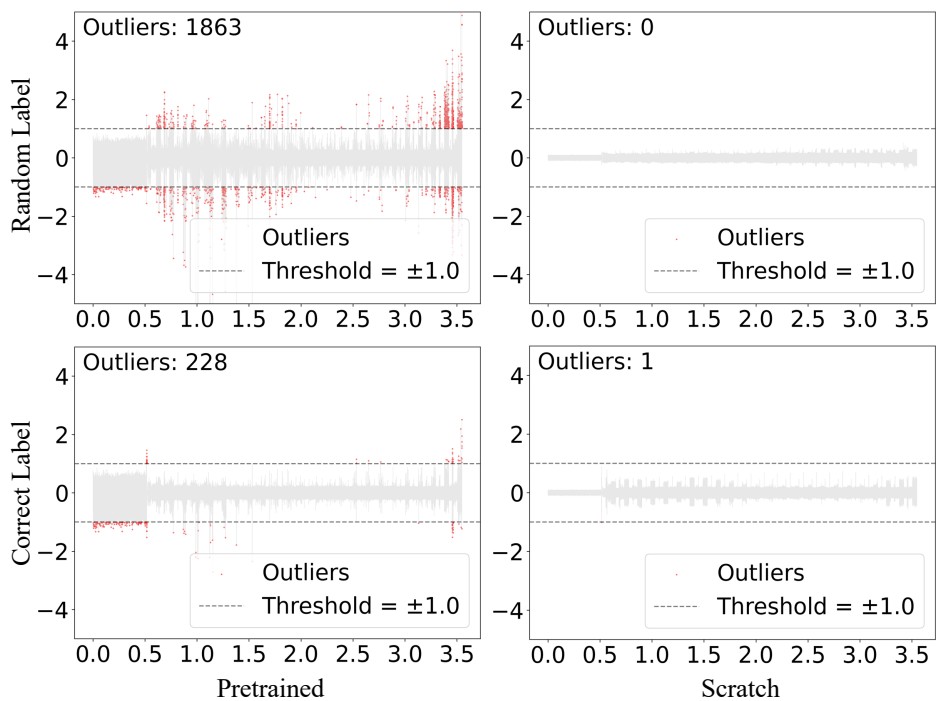

Figure 11: **Comparison of weight parameter distributions and outlier counts.** For models trained on either random or correct labels, starting from either a pretrained initialization or from scratch. Even under random labels (top row), the pretrained model continues to exhibit stronger performance than its from-scratch counterpart despite producing more high-magnitude outlier parameters. With correct labels (bottom row), the pretrained initialization still outperforms the from-scratch model, though outlier occurrences remain more frequent in the pretrained setup than in the randomly initialized one.

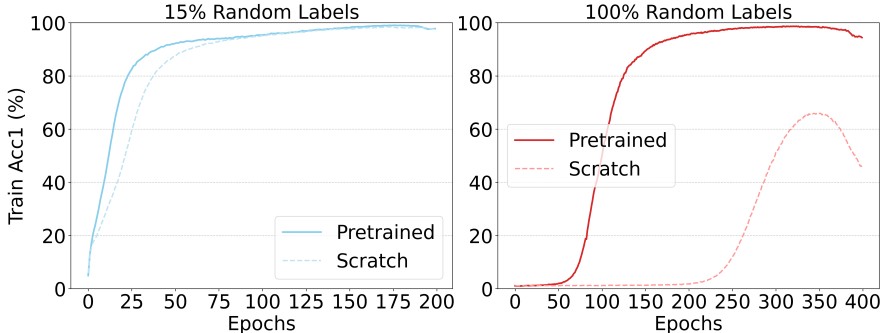

Figure 12: **Train and test acc. curves for pretrained vs. scratch GPT-2 on CIFAR-100.**

Under random labels, these differences become even more pronounced: the pretrained model exhibits a remarkably smoother and heavier-tailed distribution, demonstrating that its parameters can adapt meaningfully to visual signals despite complete label noise. This underscores our central claim that language-pretraining imparts structural biases conducive to organizing visual features, whereas the scratch-initialized model is less equipped to handle the absence of semantic guidance. Such breadth in the pretrained distribution aligns with our earlier outlier analysis in Section 3, revealing how even seemingly irrelevant label tasks may realign GPT-2's language-induced priors in ways that prove beneficial for visual understanding.

**Outliers Weights.** The results in Fig. 11 presence of numerous outlier parameters in the pretrained model under random labeling (top-left) reflects how a well-initialized feature space can amplify the impact of nonsensical supervision. Paradoxically, although random labels induce many large-magnitude weights, the pretrained model consistently surpasses the from-scratch baseline, demonstrating the robustness conferred by

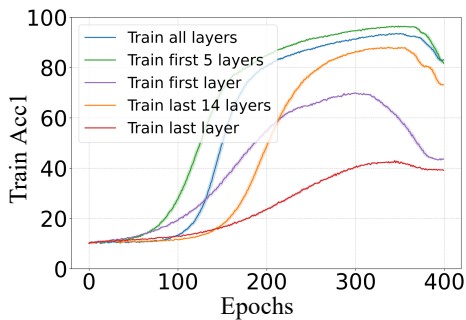

Figure 13: **More Partial Training Ablation studies on layers.**

prior training. When switching to correct labels (bottom row), the number of outliers in the pretrained model decreases substantially, showing that semantically valid supervision aligns more naturally with the pretrained parameters. Nonetheless, the pretrained model still registers slightly higher outlier counts than the scratch model, revealing that its parameter space remains more specialized—and thus more reactive—compared to the uniformly initialized parameters. Crucially, in both label scenarios, the pretrained initialization outperforms the from-scratch approach, underscoring the lasting benefits of pretrained representations even when facing adverse or unusual supervision signals.

**Train and test acc. curves for pretrained vs. scratch GPT-2 on CIFAR-100.** Fig. 12 shows training accuracy curves for pretrained versus scratch-initialized GPT-2 on CIFAR-100 under 15% and 100% random labels. Even with fully corrupted labels, the pretrained model rapidly reaches high training accuracy, whereas the from-scratch model converges more slowly and to a much lower plateau. Under partial corruption (15% random labels), the pretrained model again exhibits a faster, smoother climb in accuracy, demonstrating its enhanced robustness and more efficient adaptation compared to training from scratch.

**Image Retrieval Results on Tiny-Imagenet200.** We also demonstrate the effectiveness of language bias on image retrieval. In Table 11, the results show that Language Bias Bridge Training can surpass the model without Langugae Bias a lot on performance *e.g., 14.9% on mAP* .

Table 11: Image Retrieval Results on Tiny-ImageNet200.

| Model | mAP | Pass@1 | Recall@1 | NDCG@1 | Pass@5 | Recall@5 | NDCG@5 |
|---|---|---|---|---|---|---|---|
| Bridge Training | 72.3 | 72.1 | 72.1 | 72.1 | 71.7 | 88.1 | 81.3 |
| Language Bias Bridge Training | 87.2 | 86.6 | 86.6 | 86.6 | 86.5 | 93.5 | 90.5 |

## C   Detailed Results on Partial Training

Table 8 compares training all layers versus only the early layers under both correct-label and random-label regimes. Surprisingly, restricting updates to just the first few layers can match or even outperform full-layer training. For example, on CIFAR-10 with correct labels, training only the first 5 layers achieves 85.5%, close to 87.1% from training all layers. Under random labels, the gap is even more striking: updating only the first 5 layers yields +2.6% training accuracy than training all layers. A similar trend holds for CIFAR-100, confirming that early layers shaped by language pretraining are key to robust, transferable representations. By contrast, fully updating deeper layers can weaken these beneficial linguistic biases. Hence, partial bridge training reduces computational costs and preserves valuable language-induced structures, enabling strong cross-modality performance.

We futher conduct additional ablation studies on *partial bridge training* with random labels, further investigating which subsets of layers to fine-tune in a pretrained language model. As shown in Fig. 13 and Table 12, training only first few layers consistently converges faster and achieves higher accuracy than tuning only the last layers. Notably, when adapting random labels on CIFAR-100, updating the first five layers surpasses updating the last fourteen layers by a significant margin. This supports our claim that early layers, shaped

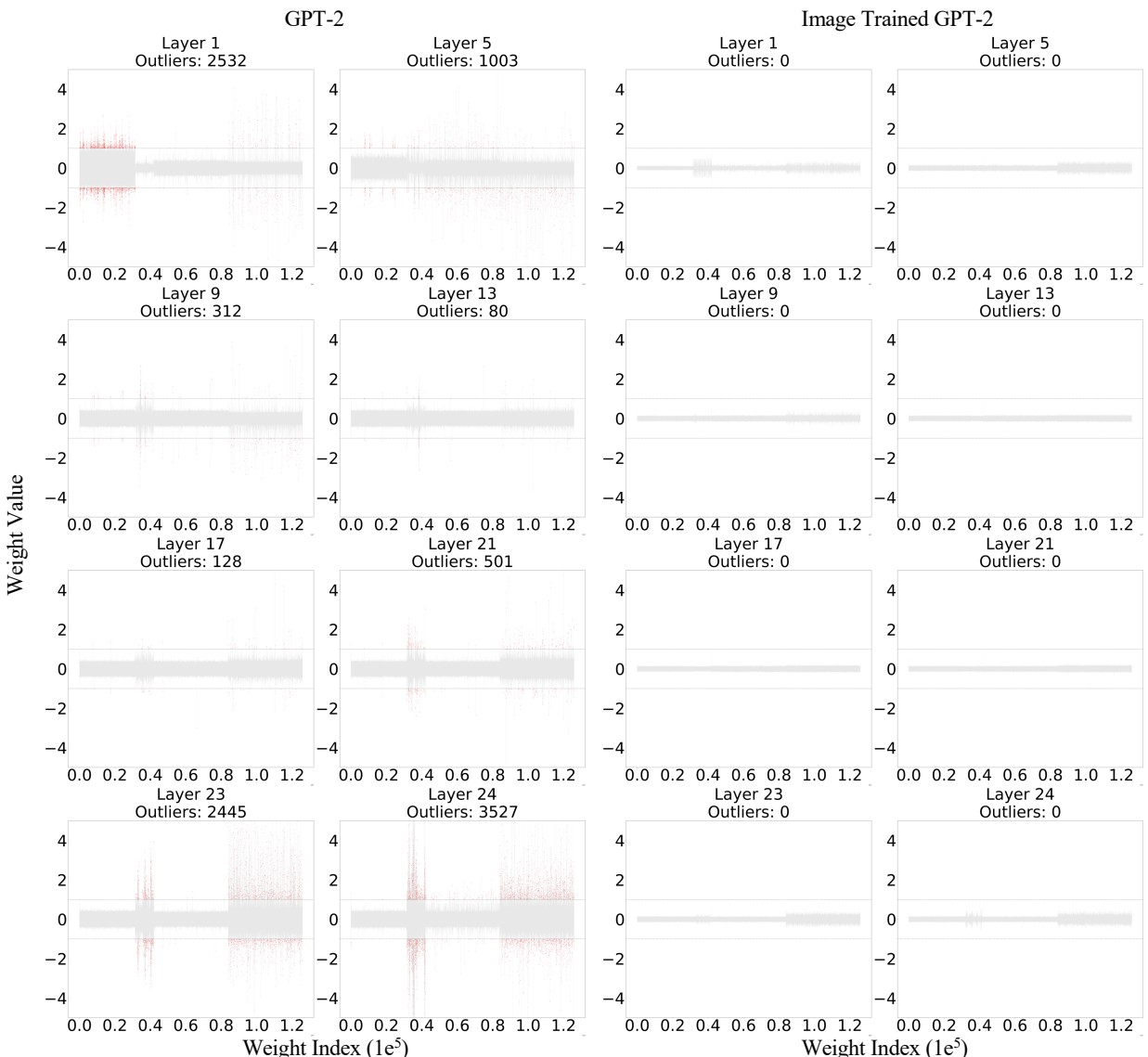

Figure 14: **Fine-grained parameter outlier and distribution comparison.**

by language pretraining, provide more flexible, general-purpose features for cross-modality adaptation—even in highly unconstrained settings such as random-label learning.

Table 12: **Extend ablation study on Random Label Partial Bridge Training.** Darker " ▮ " indicates higher Train-acc1 and Test-acc1, respectively. Note that under random-label training, the test accuracy is naturally low because the labels have no true class semantics; hence, the training accuracy is the primary indicator of model's fitting ability.

| Layers | CIFAR-10 | |
| --- | --- | --- |
| | Train-acc1 | Test-acc1 |
| Last Layer | 43.1 | 5.2 |
| Last 14 Layers | 87.9 | 6.3 |
| First Layer | 69.9 | 4.3 |
| All Layers | 93.9 | 6.3 |
| First 5 Layers | 96.5 | 6.7 |

These observations align with recent work by Skean *et al.* [46], who show that intermediate (non-final) layers often yield richer representations than the final layer. They attribute this to broader syntactic and structural information persisting in the middle of large language models. In our context, selectively training those early or intermediate layers appears to harness these more generalizable linguistic biases while keeping later layers frozen and less susceptible to overfitting nonsensical label mappings. By contrast, fine-tuning only the last few layers, where specialized or domain-specific transformations accumulate, impedes the model's ability to effectively relearn and adapt to new modalities.

Overall, these additional ablation studies underscore the practical benefits of partial bridge training: (1) it preserves the model's valuable early-layer representations that are crucial for robust convergence under noisy or random labels; (2) it helps maintain performance on real-label tasks; and (3) it reduces computational overhead by updating fewer parameters. Coupled with the findings of Skean *et al.*, our results reinforce the growing recognition that early or intermediate layers in large language models are an ideal pivot point for cross-modality adaptation, offering both flexibility and a strong inductive bias from language pretraining.

## D   Fine-grained Layer-wise Outlier Analysis

Fig. 14 extends our main analysis by providing a layer-wise comparison of weight distributions in GPT-2 and its image-trained counterpart. The left column shows that GPT-2 exhibits a substantial number of outlier parameters (highlighted in red), particularly in Layer 1, Layer 5, Layer 23, and Layer 24, where thousands of extreme values emerge. In contrast, the image-trained GPT-2 (right column) has no significant outliers across all layers, with its weight distributions appearing smoother and more compact. This contrast reinforces the idea that discrete language tokens induce sharp parameter fluctuations, whereas continuous visual signals result in more uniform parameter magnitudes.

A closer look at GPT-2's layer-wise distribution reveals a pattern: early layers (e.g., Layer 1) have the highest concentration of outliers, likely due to the initial token embedding process, while deeper layers show a gradual reduction in extreme values. However, the final layers (e.g., Layer 23 and Layer 24) again exhibit a resurgence of outliers, suggesting that language models tend to amplify certain parameter magnitudes when mapping hidden states to output space. In comparison, the vision-trained model maintains a consistently well-behaved distribution across all layers, further indicating that language pre-training inherently leads to more volatile weight dynamics. This finding underscores the challenges in cross-modal adaptation, particularly when transferring pre-trained language models to vision tasks.

## E   Compare with the Pretrain Bias from Time Series Modality

To further show that the parameter distribution learned through language pre-training can benefit vision modality, we compare the pretrained bias between language and time series using GPT-2 and chronos-t5 ( A model based on language model architectures pretrained on time series data). Table 13 shows that language pretrained bias can outperform time series pretrained bias on both correct label and random label settings across different datasets.

Table 13: Comparison between Language Pretrained and Time Series Pretrained Bias.

| Model | Imagenet100 | | CIFAR-100 | | CIFAR-10 | |
|-------|-------------|-----------|-----------|-----------|-----------|-----------|
| | Train-acc1 | Test-acc1 | Train-acc1 | Test-acc1 | Train-acc1 | Test-acc1 |
| *Correct Label* | | | | | | |
| Pretrained-GPT-2 | 86.6 | 76.2 | 99.1 | 60.4 | 99.2 | 87.1 |
| Pretrained-Chronos-t5 | 78.7 | 79.7 | 94.5 | 58.1 | 98.3 | 88.2 |
| *Random Label* | | | | | | |
| Pretrained-GPT-2 | 94.1 | 1.5 | 98.7 | 0.8 | 93.9 | 6.3 |
| Pretrained-Chronos-t5 | 80.9 | 1.3 | 92.4 | 1.1 | 87.9 | 4.3 |

## F    More Linear Probing Results

Table 14 shows the linear probing results of LBBT on CIFAR-10 and CIFAR-100, demonstrating its effectiveness.

Table 14: Linear probing image classification performance across CIFAR-10 and CIFAR-100. All models are random label bridge trained on TinyImagenet-200 first.

| Paradigm | CIFAR-10 Test-acc1 | CIFAR-100 Test-acc1 |
|---|---|---|
| w/o Bridge Training | 62.7 | 35.0 |
| Bridge Training | 74.2 | 49.1 |
| Language Bias Bridge Training | 82.3 | 56.3 |

Table 15: Quantitave analysis of loss landscape between Pretrained GPT2 and Scratch GPT2.

| Metric | Pretrained GPT2 | Scratch GPT2 |
|---|---|---|
| Eigenvalue decay rate | -11.603 | -3.691 |
| Kurtosis of eigenvalue distribution | 14.511 | -1.409 |
| Trace of Hessian | 305,404 | 20,785 |
| Spectral gap | 207,779.44 | 226.10 |
| Participation ratio | 0.336 | 0.778 |
| Noise sensitivity AUC | -0.011 | 0.003 |
| Gradient predictiveness | 0.044 | 0.830 |
| Max parameter sensitivity | 97.296 | 23.599 |
| Negative eigenvalue ratio | 0.65 | 0.35 |

## G    Quantitative Analysis of Loss Landscape.

In Section 4.3, we visualize the loss landscape of models initiated with and without language pretrained bias. Here, to further support our findings, we conduct extensive quantitative analysis comparison of the loss landscapes, as shown in Table 15.

