# OpenReview forum: "Language-Pretraining-Induced Bias: A Strong Foundation for General Vision Tasks"
_TMLR — Accepted by TMLR_

### Review · Reviewer_eanT · 2025-11-12

**Summary Of Contributions:**

This paper challenges the assumption that language and vision models have incompatible parameter spaces by pointing out that LLMs like GPT 2 have different outlier weight distributions than vision models. One of the main contributions is the Language Bias Bridge Training (LBBT), which is a simple, annotation free training where a pretrained LLM is adapted to a vision tasks using random labels. The authors show that random label training forces the model to align model parameters with the input data structure rather than the labels and adapts the language bias for visual feature extraction (demonstrated by tSNE visualizations). Another finding is that updating only the first few layers is more effective when adapting LLM with partial bridge training as it is shown to preserve high level representations.

**Audience:**

Yes

**Audience Explanation:**

The papers findings would interest the audience, as its proposed "Language Bias Bridge Training" is a simple, cheap, and annotation free way to initialize vision models using unimodal LLMs. The paper challenges existing assumptions in multimodal and transfer learning and is supported by analysis and results of model training.

**Broader Impact Concerns:**

The paper presents a general machine learning concept for cross-modal transfer learning. No immediate ethical concerns found as the work does not involve sensitive data.

**Claims And Evidence:**

Yes

**Claims Explanation:**

The paper's three main claims are mostly well-supported.
1. Claim 1 talks about the usefulness of language bias. It is evidenced in Table 1 (pretrained GPT2 outperforms a GPT2 trained from scratch on image classification obtaining +2.3% on ImageNet-1K).
2. Claim 2 focuses on random label bridge. The paper shows that pretrained models have high robustness, fitting to 100% random labels (93.9% train acc) vs scratch models (34.2%). t-SNE is also provided which shows LBBT learns separable feature clusters from random labels, while the scratch model does not.
3. Claim 3 is partial training in which authors show that training only the first 5 layers (85.5% acc) matches the performance of training all layers (87.1%)  (Figure 8 and Table 7).

The paper has a major weakness. It uses an unrealistic baseline. A GPT-2 model trained from scratch on vision might not be a competitive model. DINO would be better for this task.

**Requested Changes:**

1. The GPT-2 baseline is unrealistic. To make the paper's claims truly convincing, it is highly recommended to compare results against standard vision models like DINO.
2. The paper would be strengthened by explaining how the GPT2 is used as a backbone for dense prediction models as the section is still unclear.
3. There are some typos like “pertaining”, "emprical", "Conculsion". "initilzed" etc. Please check for language and clarity.
4. The paper would be improved by renaming or clarifying confusing terms "Bridge Training" and "Language Bias Bridge Training".
5. The authors might consider improving and moving the LLaVA experiment from Appendix E to the main paper, as it seems inline with paper's claims.
6. Another point is to discuss whether the observed benefits in LBBT or Bridge Training scale with larger LLMs (i.e 7B, 13B, 32B models).

---

> ### Author Response · Authors · 2026-02-05
> **Responses to Reviewer eanT [1/3]**
>
> We appreciate the reviewer for the detailed and constructive feedback and comments, which are definitely helpful for us to improve the quality of the paper. All the suggestions have been included and revised in the paper's updated revision. We further reply to each of your questions below:
>
> > **Q1:** A GPT-2 model trained from scratch on vision might not be a competitive model. DINO would be better for this task.
>
> As suggested, we have added additional comparisons in the table below using the representative DINO with SSL objectives under a matched pretraining budget (ImageNet-100 pretrain + identical linear probe on CIFAR-100). Under this controlled setting, LBBT achieves the best test accuracy among the compared objectives (reported in the revision). We also include a strong standard vision SSL reference (DINO ViT-L/16, 62.2% on CIFAR-100) to contextualize absolute performance. This confirms that specialized vision SSL remains superior in absolute accuracy, which does not contradict our claim: **language pretraining provides a non-trivial and transferable inductive bias**, and LBBT offers a simple, annotation-free adaptation route when starting from language-only checkpoints. We have also included the table below in the revised paper of Section 6, Table 7 under the paragraph ''Compare with other Self-Supervised Methods''.
>
> We further clarify that we do not claim scratch GPT-2 is a ''standard vision baseline" that competes with specialized vision SSL methods. Our intent is to **isolate the effect of language-pretrained initialization under a fixed Transformer architecture** (i.e., same architecture, different initialization).
>
>
> |   |   |   |
> |---|:---:|:---:|
> |Method|Train-acc1|Test-acc1|
> |**Unstructured SSL Methods**|   |   |
> |Language Bias Bridge Training|21.8|33.4|
> |Rotation Image|25.1|32.2|
> |**Structured SSL Methods**|   |   |
> |MAE|24.5|30.7|
> |SimCLR|16.2|26.2|
> |**Standard vision SSL reference**|   |   |
> |DINO (ViT-L/16)|/|62.2$^\dagger$|
> *( $^\dagger$ indicates different backbone size/ training recipe.)*
>
>
> > **R1:** **The GPT-2 baseline.** To make the paper's claims truly convincing, it is highly recommended to compare results against standard vision models like DINO.
>
> Thanks for the suggestion. Following your advice, we have included all these additional experiments (See details in our response in **Q1**). We also clarify that a scratch GPT-2 trained on vision is indeed not intended to be a competitive vision model, and we do not claim it matches specialized vision SSL systems (e.g., DINO). Our goal in using scratch GPT-2 is to provide a controlled architecture-matched counterfactual which keeps the Transformer architecture fixed and varying only the initialization, so we can isolate the effect of language-pretrained parameters on vision adaptation. We have included a more detailed discussion above, and in the revised manuscript, we further strengthen this point by adding comparisons against standard SSL objectives under a matched training budget and reporting a strong vision-SSL reference (DINO) to contextualize absolute performance (see Table 7 and related discussion in Section 6).
>
> > **R2:** **The paper would be strengthened by explaining how the GPT2 is used as a backbone for dense prediction models** as the section is still unclear.
>
> Thanks for the comment. We have carefully revised the section of our paper by adding additional details on utilizing GPT2. We also provide these details below.
>
> **Key clarification:** We do not replace DETR/Segmenter entirely with GPT-2 as the vision backbone. Instead, we exploit architectural compatibility by using pretrained GPT-2 weights to initialize specific Transformer modules inside these dense prediction models.
>
> For DETR [1], we keep the standard CNN feature backbone and detection heads, but configure the Transformer encoder/decoder with GPT‑2-matched dimensions (e.g., d=768, 12 heads, 12 layers) and copy GPT‑2 block parameters into each DETR Transformer layer (self-attention, FFN, LayerNorm). The decoder cross-attention is initialized using GPT‑2 attention weights as a reasonable warm start.
>
> For Segmenter [2], we keep the ViT encoder unchanged and initialize the mask-transformer decoder blocks using GPT‑2 pretrained block weights (attention/MLP/LayerNorm).
>
> Please have a look at our revision in Section 6 of the first two extended paragraphs in experiment ''Extend to Vision Dense Prediction Tasks'' for the concrete description of this parameter mapping and what components are (not) initialized from GPT‑2 in the dense prediction subsection to avoid ambiguity.
>
> ---
>
> References:
>
> [1] Carion, Nicolas, et al. "End-to-end object detection with transformers." European conference on computer vision. Cham: Springer International Publishing, 2020.
>
> [2] Strudel, Robin, et al. "Segmenter: Transformer for semantic segmentation." Proceedings of the IEEE/CVF international conference on computer vision. 2021.

---

> ### Author Response · Authors · 2026-02-05
> **Responses to Reviewer eanT [2/3]**
>
> > **R3:** **There are some typos** like "pertaining", "emprical", "Conculsion". "initilzed" etc. Please check for language and clarity.
>
> Thanks for pointing them out, we have fixed all the mentioned typos (''emprical'' to ''empirical'', ''Conculsion'' to ''Conclusion'', ''initilzed'' to ''initialized'') and have further gone through the whole paper carefully to correct any other typos.
>
>
>
> > **R4:** **The paper would be improved by renaming or clarifying confusing terms** "Bridge Training" and "Language Bias Bridge Training".
>
> Thanks for your kind advice, we have revised terminology and definitions to prevent confusion in Table 3. Explicitly, we define Bridge Training as an identical random-label bridge procedure (same objective, optimizer, epochs), but starting from **scratch initialization** (randomly initialized GPT-2). On the other hand, Language Bias Bridge Training means an identical procedure, but initialized from **language-pretrained GPT-2** weights.
>
> > **R5:** **The authors might consider improving and moving the LLaVA experiment from Appendix E to the main paper,** as it seems inline with paper's claims.
>
> Thanks for the suggestion. We have moved it to the main paper of Section 6 and Table 10 ("Apply Language Bias to Vision Encoder of Multimodal Large Language Model") and also extended the MLLM experiment to additional backbones and model scales. Results are summarized below (absolute score and improvement):
>
> |   |   |   |   |   |   |
> |---|---|---|---|---|---|
> |Model|LLM Backbone|SQA|TextVQA|GQA|MM-Vet|
> |Scratch-gpt2-encoder-LLaVA1.5-7B|Vicuna-7B|65.5|42.7|38.6|11.1|
> |**Pretrained-gpt2-encoder-LLaVA1.5-7B**|Vicuna-7B|**66.7 (+1.2)**|**44.2 (+1.5)**|**41.2 (+2.6)**|**18.5 (+7.4)**|
> |Scratch-gpt2-encoder-LLaVA1.6-7B|Mistral-7B|17.7|68.5|47.8|**40.9**|
> |**Pretrained-gpt2-encoder-LLaVA1.6-7B**|Mistral-7B|**21.0 (+3.3)**|**69.1 (+0.6)**|**49.3 (+1.5)**|**43.2 (+2.3)**|
> |Scratch-gpt2-encoder-LLaVA1.6-13B|Vicuna-13B|22.5|70.3|51.9|44.3|
> |**Pretrained-gpt2-encoder-LLaVA1.6-13B**|Vicuna-13B|**24.6 (+2.1)**|**71.8 (+1.5)**|**56.1 (+4.2)**|**48.7 (+4.4)**|

---

> ### Author Response · Authors · 2026-02-05
> **Responses to Reviewer eanT [3/3]**
>
> > **R6:** **Another point is to discuss whether the observed benefits in LBBT or Bridge Training scale with larger LLMs** (i.e 7B, 13B, 32B models).
>
> Thanks for your insightful suggestion. Following your comments, we have added additional scaling experiments at the most recent open-source LLMs: Qwen3-8B and Qwen3-14B (resource constraints prevent 32B; ImageNet-1K at this scale is also prohibitively expensive and time-consuming in our setup). The results show that the benefits persist at larger scales. Please check the table below and also the newly added Table 4 in our revised paper.
>
>
> |                                   |               |                       |                        |                               |
> | --------------------------------- | ------------- | --------------------- | ---------------------- | ----------------------------- |
> | Paradigm                          | LLM Backbone  | CIFAR-10 (Train-acc1) | CIFAR-100 (Train-acc1) | TinyImageNet-200 (Train-acc1) |
> | Bridge Training                   | Qwen3-8B      | 88.2                  | 84.2                   | 90.0                          |
> | **Language Bias Bridge Training** | **Qwen3-8B**  | **95.9 (+7.7)**       | **93.3 (+9.1)**        | **92.4 (+2.4)**               |
> | Bridge Training                   | Qwen3-14B     | 91.6                  | 88.3                   | 91.7                          |
> | **Language Bias Bridge Training** | **Qwen3-14B** | **97.9 (+6.3)**       | **97.7 (+9.4)**        | **94.3 (+3.4)**               |

---

### Review · Reviewer_PG3Y · 2025-11-24

**Summary Of Contributions:**

This work identifies the "outlier" parameter in vision and text foundation models; statistical analysis indicates that LLMs tend to have heavier tails while visual models provide smoother gradients (flatter distribution, fewer large outliers). Based on this observation, this paper proposes the concept of cross-modality adaptation. Further theoretical and empirical evidences are provided to show that language-pretrained parameters can help visual tasks. Then this paper proposes the LBBT (language bias bridge training); it allows to train a LLM for vision tasks.

**Audience:**

Yes

**Audience Explanation:**

This paper works on a significant and interesting topic in bridging the llm and vlm training. The proposed method seems to be new. The random label phase is impressive. As a result, I believe many audients in TMLR would be interested in knowing this method.

**Broader Impact Concerns:**

No concern.

**Claims And Evidence:**

Yes

**Claims Explanation:**

All major claims have been justified by solid evidence and theoretical analysis. For example, at the beginning of this paper, it claims that the ratio of outlier parameters in two modality models can differ significantly; later this statement is fully verified in Section 4. I have briefly scanned this paper and all statements I noticed have been validated by empirical experiments.

**Requested Changes:**

This paper has provided solid evidence to support its claim. The method is new and well-presented. The setting is clearly introduced. Therefore, I do not have further request on this paper.

One minor comment: on page 7, 5.1 Overall Proposed Formulation, there is an extra space in "paradigm , designed"; it should be "paradigm, designed".

---

> ### Author Response · Authors · 2026-02-05
> **Responses to Reviewer PG3Y**
>
> Thanks for your careful and encouraging comments. All the suggestions have been included and carefully revised in our updated paper. We further reply to your questions in the following:
>
>
> > **Minor typo:** On page 7, 5.1 Overall Proposed Formulation, there is an extra space in "paradigm, designed"; it should be "paradigm, designed".
>
> Thanks for pointing them out. We have fixed the typos on page 7, Section 5.1 (extra space in ''paradigm, designed'') in the revised version, and we also conducted an additional pass to thoroughly check for similar formatting issues and typos.

---

> > ### Comment · Reviewer_PG3Y · 2026-02-06
> >
> > Thanks for the comment. Everything looks good for me; other reviews and rebuttals do not raise additional concerns on my side. I will recommend accepting.
> >
> > A minor typo: On page 2, "Our contributions are threefold: [...] We challenge that, ..." . The item (1) appears to be missing in the revised version.

---

> > > ### Author Response · Authors · 2026-02-06
> > >
> > > Thanks for the kind comments and for pointing it out. We have corrected it in the updated version.

---

### Review · Reviewer_sCfh · 2026-01-21

**Summary Of Contributions:**

This paper studies the empirical properties of a simple cross-modal fine-tuning procedure, which they call "Language Bias Bridge Training" (LBBT). LBBT consists of taking a pretrained LLM, then (1) fine-tuning it to predict random labels on image data, then (2) fine-tuning it on labelled data; possibly while keeping the parameters of some layers frozen. Theoretically, they show that, consistent with prior results, random label training induces a form of alignment between the input data and the first-layer weights of a randomly initialized network.

Empirically, they consider tasks including CIFAR-10, CIFAR-100, ImageNet-1K, and TinyImageNet-200, along with GPT-2 models and the LLAMA3.2-1B model. They find that:
1. Across different percentages of random versus true labels, using a pre-trained LLM improves performance over training the same architecture from scratch [Section 4.2].
2. Using a "language bias" (i.e., a pre-trained LLM) induces a better initial loss landscape according to several metrics (e.g., higher Hessian trace, indicating larger curvature and thus greater potential for decreasing loss), both for random labels and for true labels [Section 4.3].
3. For random labels, the language bias also leads to learned representations which are more separable [Section 4.4].
4. Restricting updates to the first few layers can match our outperform training all parameters [Section 6].
5. Similarly improvements hold for other vision tasks, e.g. object detection (COCO 2017) and semantic segmentation (AED20K and Pascal Context).

## Strengths
1. **Good experimental coverage:** I appreciated that the experiments were performed across several tasks and datasets, and that several different aspects were investigated - the experiments did not just look at performance, but also at explanatory phenomenon like properties of the optimization landscape.
2. **Clear distinction between multi-modal and multi-domain:** In my experience, these two terms are often conflated, and it is nice that the authors are carefully to distinguish the two. Typical work on domain adaptation considers cases where the source and target tasks take inputs in the same mathematical space (e.g. natural vs. cartoon images), whereas this paper allows those spaces to be different, as they highlight in Definition 3.1

## Weaknesses
1. **Somewhat weak motivation:** There is not a particular motivating example, e.g. a setting where there is a large amount of text data but a small amount of image data. I understand that the value of the paper may be of more scientific interest rather than practical interest, but as a more pragmatic reader, this lack of motivation was distracting. In particular, the experiments consider a 355M parameter model, which seems extremely large for the four image datasets (e.g. a 20-layer ResNet with 300K parameters can achieve better training accuracy on CIFAR-10 than what’s show in Table 3).
2. **Poor clarity about some details:** Additional clarification is needed in several places, see “Requested Changes”.

**Audience:**

Yes

**Audience Explanation:**

The empirical findings build on previous work, and have scientific merit. Learning / aligning across modalities is generally an important area with lots of interest across a wide audience, so I could see some of TMLR's audience being interested in these findings.

**Claims And Evidence:**

Yes

**Claims Explanation:**

The main claim is that "Language Bias Bridge Training", even with random labels, helps LLM parameters adapt to vision tasks, i.e., that there is some performance benefit to the cross-modality transfer from language to vision. The experiments support these claims across a few different tasks, datasets, and architectures, with appropriate comparisons.

**Requested Changes:**

## Major Changes (critical to securing my recommendation)
1. **Clarify how images are input:** A KEY detail which I should have understood, but did not, was how the images are input into GPT-2. Indeed, this detail is key to understanding how the multi-modality framework is supposed to work at all, since the pre-trained model is supposed to take inputs in a different mathematical space. Please clarify in detail.
2. **Clarify hyperparameter selection details:** In Section 6, you say that the hyperparameters (e.g. weight decay and learning rate) are consistent, but not how they are picked. In particular, the choice may be unintentionally favorable to your desired results; it would be more scientifically rigorous to confirm that the phenomena hold across a range of hyperparameters.
3. **Theorem 4.1 discussion and comparison to existing results:** To the best of my understanding, compared to previous results, Theorem 4.1 only differs in that it is more architecture-agnostic, is this correct? It would be good to add a more explicit discussion of how the theorem compares to existing results. Moreover, the theorem assumes randomly-initialized first layer weights, which does not hold when the weights are initialized from a pre-trained model. It would be good to discuss this mismatch between the theorem’s conditions and the actual method.
4. **Discussion of “incompatibility”:** This paper mentions that there is a “common notion that language and vision parameter spaces are categorically incomparable”. For some readers, that notion might resonate. However, this notion did not seem so “common” to me, e.g. ViT have been a successful application of language-based architectures (“parameter spaces”) to vision, and diffusion language models have been a successful application in the reverse direction. Moreover, work on representation alignment (e.g. as detailed in [1]) points to more general phenomenon of cross-modality alignment. Ultimately, it may be better to remove/modify the idea that “incompatibility” is something that many people take for granted; many people in my community would not agree with this point.

[1] Huh et al. (2024), “The Platonic Representation Hypothesis”

## Minor Changes (would simply strengthen the work)
1. **Motivating example:** Even if the example is not explored in the experiments, it would be helpful to add a motivating example, i.e., a setting where there is only a small amount of image data, but a large amount of text data. Otherwise, it should be emphasized that the results are primarily of scientific, rather than practical, interest.
2. **Clarify definition of “task”:** In Definition 3.1, please also define the notion of “learning task” and how it is “derived”, e.g. in “$t_S$ is the learning task derived from $Y_S$”.
3. **Clarify loss landscape:** In Figure 4, please explain what are the X and Y axes of the image (since the parameter space is high-dimensional, this must be a cross-section?).
4. **Clarify neuron activation ratio:** In Section 6, please clarify the meaning of “the percentage of image samples that trigger neuron activations…”, i.e., is there a threshold that determines whether a neuron counts as “activated”?
5. **Clarify “standard” bridge  training:** Table 3 compares LBBT against “standard Bridge Training”, but it’s not clear to me what you are referring to.

---

> ### Author Response · Authors · 2026-02-05
> **Responses to Reviewer sCfh [1/5]**
>
> Thank you for the thorough and constructive review. We have carefully revised the manuscript to include all the mentioned suggestions and changes. We further reply to each of your questions as follows:
> > **W1: Somewhat weak motivation:** There is not a particular motivating example, e.g. a setting where there is a large amount of text data but a small amount of image data. I understand that the value of the paper may be of more scientific interest rather than practical interest, but as a more pragmatic reader, this lack of motivation was distracting. In particular, the experiments consider a 355M parameter model, which seems extremely large for the four image datasets (e.g. a 20-layer ResNet with 300K parameters can achieve better training accuracy on CIFAR-10 than what’s show in Table 3).
>
> Thanks for the suggestion. In our revision, we have added an explicit motivating example in the _Introduction Section_ and also a pragmatic setting where there is abundant text but limited image supervision: domain text (reports, manuals, logs, or expert notes) is plentiful, while collecting and labeling domain images is expensive or slow, yet unlabeled images may still be available. In such cases, one may start from a language-pretrained LLM checkpoint and adapt it into a vision-capable backbone with minimal or weak supervision, our bridge stage is designed exactly for this ''language-first, vision-scarce'' regime and does not rely on paired image-text data.
>
> Regarding the model size, yes, a GPT2-like model architecture is not the most practical architecture for _Image Classification_, and we do not claim competitive supervised accuracy against vision-specialized CNN/ViT recipes (e.g., ResNets). We clarify that we use the 355M setting primarily because it corresponds to a standard, widely-used GPT-2 checkpoint and enables a controlled study that isolates the effect of language-pretrained initialization under a fixed architecture and training protocol. Our purpose is not comparing GPT-2 vs. ResNet, rather, we compare the same architecture trained from scratch versus with language pretraining, and our conclusion concerns the effect of this kind of prior initialization.
>
> > **W2: Clarity about some details:** Additional clarification is needed in several places, see “Requested Changes”.
>
> Thanks for the comment and detailed suggestions. We have carefully gone through all the "Requested Changes" and revised the manuscript accordingly to improve clarity. Please check out our updated version of the paper and our following detailed responses for the clarifications.
>
> > **Major** **1****:** **Clarify how images are input:** A KEY detail which I should have understood, but did not, was how the images are input into GPT-2. Indeed, this detail is key to understanding how the multi-modality framework is supposed to work at all, since the pre-trained model is supposed to take inputs in a different mathematical space. Please clarify in detail.
>
> Thanks for pointing this out. We have added an explicit clarification in the revised paper. Please see Section 6's two new paragraphs. For your convenience, we also provide an explicit and clear step-by-step description below:
>
> Although the backbone is GPT-2, we do not feed raw pixels into GPT-2's text-token embedding table. Instead, we introduce a **ViT-style patch embedding** that maps an image into a sequence of continuous embeddings whose hidden size matches GPT-2. Then we apply the standard GPT-2 Transformer blocks (initialized from the language-pretrained GPT-2 weights). Concretely, given an image $X$ ($H \times W\times 3$), we:
> 1. **Patchify + projection:** split the image into non-overlapping ($P \times P$) patches and then each patch is projected by a learnable linear/conv projection into a d-dimensional embedding (with d matching GPT-2 hidden size).
>
> 2. **Add positional encoding:** add learnable positional embeddings to form the input token sequence.
>
> 3. **Process with GPT-2 blocks:** feed the sequence through GPT-2 Transformer blocks which are initialized from the pretrained language GPT-2 checkpoint.
>
> 4. **Task head:** for classification, we pool the final token representations and apply a linear classifier trained with cross-entropy.
>
> **To answer** **why this resolves the "mathematical space mismatch''**, the _only_ modality-specific interface is the patch embedding layer, once mapped into the shared d-dimensional embedding space, the subsequent Transformer computation is standard and can be initialized from language pretraining.
>
> **For the MLLM setting**, we also clarify the alignment stage used by LLaVA: vision features are projected into the LLM embedding space by a small MLP trained on image-text pairs, while other modules are frozen. In our experiments, we change only the vision encoder (CLIP → GPT-2-style transformer blocks) and follow the original LLaVA training recipe; the alignment mechanism remains the same.

---

> ### Author Response · Authors · 2026-02-05
> **Responses to Reviewer sCfh [2/5]**
>
> > **Major** **2****:** **Clarify hyperparameter selection details:** In Section 6, you say that the hyperparameters (e.g. weight decay and learning rate) are consistent, but not how they are picked. In particular, the choice may be unintentionally favorable to your desired results; it would be more scientifically rigorous to confirm that the phenomena hold across a range of hyperparameters.
>
> Thanks for the comment. In the revised manuscript, we have added a learning rate _vs._ weight decay sensitivity study and reported the full grid in Table 3 (and in the table below). We totally agree that optimization hyperparameters (learning rate and weight decay) can affect absolute performance.
>
> We also clarify our protocol: we run a small sweep only to choose a single hyperparameter setting for the baseline (without LBBT) using a method-agnostic selection criterion. We then freeze this choice and apply the exact same optimizer, schedule, compute budget, and LR/WD across all variants (including LBBT), preventing any inadvertent tuning in favor of LBBT.
>
> Overall, while absolute accuracy varies across hyperparameter settings, LBBT consistently outperforms training without LBBT for all tested learning rate and weight decay combinations. We include the complete sweep grid and results in a new subsection, "Hyperparameter selection and robustness" in Section 6, and we also summarize them in the table below:
>
> |   |   |   |   |
> |---|:---:|:---:|:---:|
> |lr \ wd|wd = 0.05|wd = 0.1|wd = 0.2|
> |lr = 5e-4|15.1 / 31.0 (+15.9)|10.5 / 30.3 (+19.8)|10.3 / 23.5 (+13.2)|
> |lr = 3e-4|17.2 / 32.4 (+15.2)|13.9 / 31.1 (+17.2)|12.6 / 24.8 (+12.2)|
> |lr = 1e-3|17.4 / 35.0 (+17.6)|16.6 / 33.3 (+16.7)|13.7 / 26.8 (+13.1)|

---

> ### Author Response · Authors · 2026-02-05
> **Responses to Reviewer sCfh [3/5]**
>
> > **Major 3:** **Theorem 4.1 discussion and comparison to existing results:** To the best of my understanding, compared to previous results, Theorem 4.1 only differs in that it is more architecture-agnostic, is this correct? It would be good to add a more explicit discussion of how the theorem compares to existing results. Moreover, the theorem assumes randomly-initialized first layer weights, which does not hold when the weights are initialized from a pre-trained model. It would be good to discuss this mismatch between the theorem's conditions and the actual method.
>
> Thanks for the comment. We clarify that (i) how the theorem relates to prior "random labels" analyses and (ii) how assumptions match the actual method as below. We have also carefully revised the relevant part of the paper.
>
> **(i) Comparison to existing results.** Our theorem is consistent with prior findings that networks trained on random labels can still learn input structure. For example, one prior work has established that over-parameterized networks can fit completely random labels (and even random inputs), indicating that optimization can succeed even without label semantics[1]. Beyond this memorization fact, more recent analyses have investigated what structure is learned in this regime. In particular, Maennel et al. [2] analytically show that, under isotropic initialization, training with random labels aligns the principal components (eigenvectors) of the first-layer weight covariance with those of the data covariance, and empirically observe an approximately one-dimensional spectral transfer mapping from the data spectrum to the weight spectrum. Complementarily, studies of memorization dynamics show that networks tend to learn simple / shared patterns before memorizing noise, consistent with early layers capturing input structure even when labels are random [3][4].
>
> Our goal is not to claim a qualitatively new phenomenon, but to present a formulation **tailored to cross-modality adaptation**: we formalize how random-label training induces **covariance alignment** between the _input-facing learnable interface_ and the data covariance, including (a) eigenvector alignment and (b) an empirically smooth spectral mapping (data-spectrum → weight-spectrum). In the revision, we add an explicit paragraph after Theorem 4.1 describing what is shared with prior analyses and what is novel in our viewpoint (covariance-alignment mechanism + spectral interpretation + direct connection to our "bridge stage").
>
> **(ii) "Random first layer" vs. pretrained initialization.** Importantly, the theorem's assumption that matters is the initialization of the **input-facing layer**. In our vision adaptation, this layer is the newly introduced **patch embedding / input projection**, which is **randomly initialized** (isotropic Xavier/normal), because the original GPT-2 token embedding cannot intake pixels. The pretrained GPT-2 weights are loaded into the subsequent Transformer blocks. Therefore, the theorem's key condition matches our actual implementation of image inputs and provides a mechanistic explanation for why the random-label bridge stage can align the input projection to the image distribution even without semantic labels. We also clarify that the theorem should be interpreted as describing an alignment _dynamics_ mechanism; empirically we observe the same covariance-alignment trend during the random-label stage
>
> ---
>
> References:
>
> [1] Zhang, Chiyuan, et al. "Understanding deep learning requires rethinking generalization." _arXiv preprint arXiv:1611.03530_ (2016).
>
> [2] Maennel, Hartmut, et al. "What do neural networks learn when trained with random labels?." _Advances in Neural Information Processing Systems_ 33 (2020): 19693-19704.
>
> [3] Arpit, Devansh, et al. "A closer look at memorization in deep networks." _International conference on machine learning_. PMLR, 2017.
>
> [4] Anagnostidis, Sotiris, et al. "The curious case of benign memorization." _arXiv preprint arXiv:2210.14019_ (2022).

---

> ### Author Response · Authors · 2026-02-05
> **Responses to Reviewer sCfh [4/5]**
>
> > **Major** **4****:** **Discussion of "incompatibility":** This paper mentions that there is a "common notion that language and vision parameter spaces are categorically incomparable". For some readers, that notion might resonate. However, this notion did not seem so "common" to me, e.g., ViT has been a successful application of language-based architectures ("parameter spaces") to vision, and diffusion language models have been a successful application in the reverse direction. Moreover, work on representation alignment (e.g., as detailed in [1]) points to more general phenomenon of cross-modality alignment. Ultimately, it may be better to remove/modify the idea that "incompatibility" is something that many people take for granted; many people in my community would not agree with this point.
> >
> > [1] Huh et al. (2024), "The Platonic Representation Hypothesis"
>
>
> We appreciate the feedback. Following your suggestion, we have removed and revised the statements that implied the claim that "language and vision parameter spaces are categorically incomparable" is a commonly held belief. Instead, we now emphasize a narrower and more precise point as follows:
>
> (i) **Architectural compatibility is well established** (e.g., Transformers work in vision; cross-directional ideas exist). (ii) The open question we study is **whether language-pretrained weights provide a measurable advantage** when adapting to vision under _minimal or weak cross-modal supervision_, especially in the random-label bridge stage where label semantics are absent. Specifically, we have modified the previous strong words in Section Introduction, Section 4, and Section Conclusion of the new paper revision.
>
> We appreciate the pointer to representation alignment perspectives. And also position our work relative to representation alignment perspectives in the revised Related Work section (e.g., Platonic representation hypothesis [1]). Our work is complementary: rather than primarily measuring representational convergence as scale increases, we study a concrete adaptation procedure and its training dynamics (random-label bridge + partial-layer updates).
>
> [1] Huh et al. (2024), "The Platonic Representation Hypothesis".
>
> > **Minor** **1****:** **Motivating example:** Even if the example is not explored in the experiments, it would be helpful to add a motivating example, i.e., a setting where there is only a small amount of image data, but a large amount of text data. Otherwise, it should be emphasized that the results are primarily of scientific, rather than practical, interest.
>
> We have added an explicit motivating scenario in the Introduction: many specialized domains have abundant text but scarce labeled images (e.g., industrial inspection, medical imaging, remote sensing). In such settings, one can (a) leverage language-only pretraining implicitly via an LLM checkpoint, (b) adapt to domain images via an annotation-free random-label bridge stage (using _unlabeled_ images), and then (c) fine-tune with the limited labeled images available. We also clarify that our goal is **not** to claim GPT-2-scale Transformers are the best practical choice for CIFAR, but to **isolate the effect of language-pretrained initialization under controlled architecture.**
>
> > **Minor** **2****:** **Clarify definition of "task":** In Definition 3.1, please also define the notion of "learning task" and how it is "derived", e.g. in "t_S is the learning task derived from Y_S".
>
> Thanks for pointing it out. We clarify Definition 3.1 by explicitly defining a learning task as the supervised risk-minimization problem induced by the label space and the data distribution / conditional, together with a loss function. This removes ambiguity in ''derived from Y_S''.
>
> > **Minor** **3****:** **Clarify loss landscape:** In Figure 4, please explain what are the X and Y axes of the image (since the parameter space is high-dimensional, this must be a cross-section?).
>
> Thanks for the suggestion and careful review, yes, the loss is defined over a high-dimensional parameter space, so Fig. 4 visualizes a **2D cross-section**. Concretely, we plot:
>
> $\mathcal{L}(\theta_0 + \alpha d_1 + \beta d_2),$
>
> where $\theta_0$ is the initialization,  and $\theta_T$ is the final trained model, and $d_1 = \theta_T - \theta_0$ is the **training direction**. We sample a random direction $d_2$ such that $d_2 \perp d_1$, and apply **per-layer normalization** to both $d_1$ and $d_2$ to avoid domination by layers with larger parameter scales. The **X-axis** and **Y-axis** correspond to the scalar coefficients $\alpha$ and $\beta$, respectively (the height/color indicates the loss value on this 2D slice). This explicit definition has been added to the caption of Fig. 4 in the revised manuscript.

---

> ### Author Response · Authors · 2026-02-05
> **Responses to Reviewer sCfh [5/5]**
>
> > **Minor 4:** **Clarify neuron activation ratio:** In Section 6, please clarify the meaning of "the percentage of image samples that trigger neuron activations…"", i.e., is there a threshold that determines whether a neuron counts as "activated"?
>
> Thanks for the comment. We clarify that activation is defined with an explicit threshold. For each Transformer block, we record the MLP activations after the nonlinearity (post-GELU) for each image. For a given neuron, we first average its activation over all visual tokens of that image to obtain a single scalar value per image. We then count the neuron as activated by that image if this token-averaged activation is strictly larger than 0 (threshold = 0). The reported neuron activation ratio at a layer is computed as the fraction of (image, neuron) pairs that are activated under this rule (equivalently, it can be interpreted as the neuron-averaged percentage of images that activate a neuron). We added this clarification both in Section 6 and in the caption of Fig. 7 in the revised manuscript.
>
>
>
> > **Minor** **5****:** **Clarify "standard" bridge training:** Table 3 compares LBBT against "standard Bridge Training", but it's not clear to me what you are referring to.
>
> Thanks for pointing this out, we have clarified the definition in the revised paper of Table 3 and also provide here:
>
> **Bridge Training (BT)**: identical random-label bridge procedure (same objective, optimizer, epochs), but starting from **scratch initialization** (randomly initialized GPT-2).
>
> **Language Bias Bridge Training (LBBT)**: identical procedure, but initialized from **language-pretrained GPT-2** weights.

---

### Author Response · Authors · 2026-02-23
**A summary of our rebuttal and a revised version of our manuscript have been updated**

We extend our gratitude to all the reviewers for your invaluable comments. We have diligently prepared a thorough response to address all your concerns and updated the manuscript accordingly to your suggestions.

We are encouraged by the positive comments from reviewers, including good experimental coverage, all major claims have been justified by solid evidence and theoretical analysis, claim 1 is evidenced in Table 1, this paper has provided solid evidence to support its claim [Reviewer **sCfh**, **PG3Y**, **eanT**], clear distinction between multi-modal and multi-domain [Reviewer **sCfh**], this paper works on a significant and interesting topic in bridging the llm and vlm training [Reviewer **PG3Y**], the method is new and well-presented, the setting is clearly introduced, the random label phase is impressive, its proposed "Language Bias Bridge Training" is a simple, cheap, and annotation free way to initialize vision models using unimodal LLMs [Reviewer **PG3Y**, **eanT**], I believe many audients in TMLR would be interested in knowing this method, the papers findings would interest the audience [Reviewer **PG3Y**, **eanT**].

We also thank the reviewers for constructive comments, such as somewhat weak motivation [Reviewer **sCfh**], minor typo [Reviewer **PG3Y**], vision model baseline [Reviewer **eanT**]. We have accommodated all of the comments and suggested extra experiments in our revised manuscript, where all the revised contents are highlighted in $\textcolor{orange}{\text{orange}}$ color.

Detailed responses to each reviewer's questions, concerns, and requested changes are provided in the following replies. We are eager for you to explore our detailed responses. We always welcome further discussions and comments, and your insights are crucial to refining our paper.

---

### Decision · Action_Editor_Q58h · 2026-03-23

**Recommendation:** Accept as is

**Audience:**

Yes

**Audience Explanation:**

Overall pre-training is a topic of interest to a lot of the TMLR community; as such I think the paper would be of interest.

**Claims And Evidence:**

Yes

**Claims Explanation:**

All reviewers unanimously agree that the paper provides sufficient evidence for its claims.